# Grounding Multi-Hop Reasoning in Structural Causal Models via Group Relative Policy Optimization

Yunhan Bu [* 1 2]   Quan Zhang [* 3]   Zhang Huaping [1 2]   Guotong Geng [3]   Chunxiao Gao [1]   Askar Hamdulla [2]
Juan Wang [1]   Qiuchi Li [1]   Baohua Zhang [1]   Yunbo Cao [3]   Zhunchen Luo [3]   Shuai Lei [3]

## Abstract

Multi-Hop Fact Verification requires complex reasoning across disparate evidence, posing significant challenges for Large Language Models , which may suffer from hallucinations and fractured logical chains. Existing methods, while improving transparency via Chain-of-Thought , often lack explicit modeling of the structural dependencies between evidence and claims. In this work, we introduce an SCM-inspired framework that grounds reasoning in explicit directed dependency graphs, treating verification as a constructive structural reasoning process rather than full causal inference with interventions or counterfactual semantics. We empirically identify an "inverted U-shaped" correlation between reasoning-chain length and accuracy, revealing that excessive structural complexity can degrade performance. To address this, we propose a rule-based reinforcement learning strategy using Group Relative Policy Optimization. This approach dynamically optimizes the trade-off between structural depth and conciseness. Extensive experiments on HoVer and EX-FEVER demonstrate that our SCM-GRPO framework outperforms strong baselines while producing more traceable reasoning structures for complex fact verification.

## 1. Introduction

Automated fact verification has emerged as a critical mechanism for mitigating the proliferation of misinformation, particularly given the exponential growth of online content (Guo et al., 2022). Within this domain, **Multi-Hop Fact Verification (MHFV)** stands as a paramount challenge. Unlike single-hop verification which relies on direct matching, MHFV requires models to retrieve discrete evidence and synthesize coherent reasoning chains across multiple sources. This process demands cross-document semantic linkage, the resolution of conflicting information, and the inference of implicit facts, imposing stringent requirements on a model's logical reasoning capabilities (Cai et al., 2025b; Yang et al., 2026a).

While Large Language Models (LLMs) have progressively deepened their integration into MHFV (Achiam et al., 2023; Yang et al., 2025; Llama Team, AI @ Meta, 2024), significant limitations remain. Current approaches, such as ProgramFC (Pan et al., 2023) or Supervised Fine-Tuning (SFT) methods (Wei et al., 2022; Wang et al., 2023), predominantly focus on learning end-to-end mappings from claims to labels. These methods do not explicitly model the intrinsic logical dependencies connecting evidence to claims. Consequently, in complex scenarios, such models remain prone to fractured logical chains and erroneous evidence attribution, leading to hallucinations (Huang et al., 2023; Zhang et al., 2023). Although structured frameworks such as Graph Neural Networks (GNNs) have improved evidence integration, a critical gap persists: the lack of **explicit modeling of directed structural dependencies**, which is important for reliable and interpretable verification (Feder et al., 2022; Geiger et al., 2021; Shi et al., 2025).

To bridge this gap, we introduce a **Structural Causal Model (SCM)-inspired structural dependency framework** to characterize the reasoning process. The framework models dependencies via a Directed Acyclic Graph (DAG) composed of exogenous variables (evidence) and endogenous variables (intermediate inferences and final verdicts) (Cai et al., 2025a). In the decoding phase, we embed a **Chain-of-Thought (CoT)** mechanism guided by this structural graph to improve logical grounding (Shi et al., 2024). However, we identify a critical trade-off: unconstrained SCM-CoT generation can produce excessively granular reasoning chains. This structural complexity introduces noise, which

---

[*]Equal contribution [1]Beijing Institute of Technology, Beijing, China [2]School of Computer Science and Technology, Xinjiang University, Urumqi, China [3]Military Science Information Research Center, Academy of Military Science, Beijing, China. Correspondence to: Juan Wang <wangjuan99@bit.edu.cn>, Guotong Geng <ggtong@163.com>, Askar Hamdulla <askar@xju.edu.cn>.

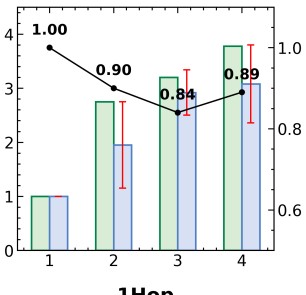 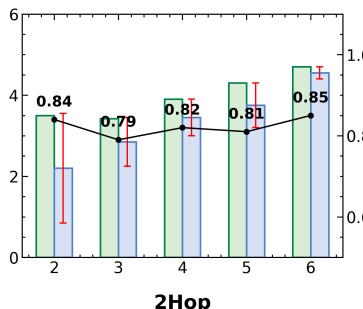 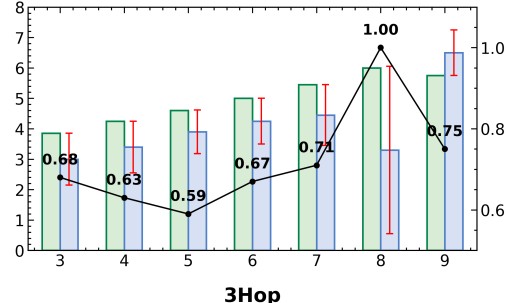

**1Hop** **2Hop** **3Hop**

*Figure 1.* Relationship between SCM-CoT structural variables and verification accuracy. Green bars denote exogenous variables, blue bars denote endogenous variables, and error bars indicate their difference; the line plot shows verification accuracy. We approximate reasoning-chain length by the number of generated structural variables. Accuracy is positively correlated with the exogenous–endogenous variable difference and is also affected by the consistency between hop count and reasoning-chain length: shorter chains are preferable for low-hop cases to avoid over-reasoning, whereas overly short chains for high-hop cases or overly long chains for low-hop cases tend to reduce accuracy.

may paradoxically impair verification accuracy (Guan et al., 2025b). We emphasize that our SCM formulation is used as a structural dependency abstraction for evidence-grounded reasoning, rather than full causal inference involving interventions, counterfactuals, or do-calculus.

We investigate this phenomenon by modeling SCM-CoT with Qwen3-30B-A3B on the EX-FEVER dataset. As illustrated in Figure 1, we reveal an **inverted U-shaped correlation** between chain length and accuracy: performance improves with length initially but degrades rapidly beyond an optimal threshold. Furthermore, we observe a positive correlation between accuracy and the ratio of evidence to inference steps. Based on these insights, we propose a **rule-based reinforcement learning (RL)** strategy utilizing Group Relative Policy Optimization (GRPO) (Shao et al., 2024). This strategy dynamically optimizes the trade-off between structural depth and conciseness, suppressing unsupported reasoning steps while maintaining logical integrity.

The main contributions of this paper are:

- **SCM-Inspired Structural Reasoning Framework:** We introduce an SCM-inspired dependency graph for MHFV, providing an interpretable foundation that explicitly models directed structural dependencies between evidence and claims.

- **Empirical Insight and RL Optimization:** We reveal an inverted U-shaped relationship between reasoning length and accuracy and propose a GRPO-based RL strategy to adaptively constrain structural complexity, balancing rigor with parsimony.

- **Strong Empirical Performance and Traceability:** Our SCM-GRPO framework achieves strong performance on public benchmarks (HoVer and EX-FEVER), while producing reasoning chains with explicit graph-based traceability.

## 2. Related Work

### 2.1. Multi-Hop Fact Verification

MHFV requires aggregating discrete evidence to judge complex claims (Li et al., 2026b; Tong et al., 2025; Li et al., 2026c; Huang et al., 2025). Early **"Decompose-and-Verify"** methods (Min et al., 2023; Zhong et al., 2023) often fail when logical connections depend on implicit facts (Li et al., 2025d; Chen et al., 2025a; 2026). While recent structured frameworks using GNNs (Besta et al., 2024; Zhao et al., 2023) or symbolic programs (Pan et al., 2023; Chen et al., 2023) improve evidence integration, they primarily focus on explicit associations (Hu et al., 2026; Chen et al., 2025b; Fu et al., 2025). They lack deep modeling of intrinsic **logical dependency mechanisms** between evidence and claims, which is essential for reliable and traceable verification (Geiger et al., 2021).

### 2.2. Supervised Fine-Tuning for LLMs

SFT-based methods fine-tune LLMs on *(claim, evidence, explanation)* data to generate verdicts and reasoning chains (Lightman et al., 2024; Lyu et al., 2023; Li et al., 2025a; 2026a). While leveraging LLMs' semantic capabilities (Jia et al., 2026; Gu et al., 2025; Yao et al., 2025; Yang et al., 2026b), this end-to-end paradigm captures statistical correlations rather than rigorous logic (Cai et al., 2025b). Consequently, models often produce superficially fluent but logically fragile explanations. In multi-hop scenarios, this susceptibility to **hallucinated reasoning** and attribution errors significantly undermines trustworthiness (Manakul et al., 2023; Elaraby et al., 2023).

### 2.3. Reinforcement Learning for LLMs

RL optimizes reasoning coherence via feedback signals (Ouyang et al., 2022; Rafailov et al., 2023; Guan et al.,

2025a), with recent methods like GRPO (Shao et al., 2024) further stabilizing training. However, applying general-purpose RL to high-stakes fact verification is challenging due to the difficulty of designing precise reward functions (Li et al., 2025c; 2023). Furthermore, without explicit structural constraints, RL-optimized models may still generate **structurally redundant** or unfocused chains (Shi et al., 2025; 2024), failing to fundamentally mitigate unsupported intermediate reasoning.

## 3. Methodology

In this section, we present our proposed framework for reliable multi-hop fact verification. As illustrated in Figure 2, the overall training pipeline is composed of two sequential stages.

### 3.1. Task Definition and Formalization Framework

This work focuses on MHFV, designed to automatically scrutinize complex claims that cannot be directly verified through a single piece of evidence. The overall workflow is illustrated in Figure 2.

Specifically, given a claim $x$ and a collection of relevant evidence documents $\mathcal{D}$, the objective is to predict the final veracity label $y$ (i.e., *Supported* or *Refuted*). The core challenge lies in the fact that the verdict $y$ often cannot be derived solely from any individual piece of evidence. Instead, it necessitates multi-step retrieval and reasoning, where the model must synthesize information from diverse sources to iteratively construct a complete and coherent logical chain.

To formalize this reasoning procedure, we model MHFV as a sequential decision-making process. Let the entire reasoning process consist of $N$ steps, denoted as the sequence $\mathcal{C} = \{(\tau_t, \alpha_t, o_t)\}_{t=1}^{N}$, where $\tau_t$ represents the reasoning state at step $t$, $\alpha_t$ denotes the action taken by the model in the current state (i.e., intermediate inference), and $o_t$ corresponds to the observation returned by the environment (typically the reasoning conclusion derived from the previous step). Through this step-by-step progression, the model is required to synthesize discrete pieces of evidence to ultimately formulate a globally consistent verification conclusion.

### 3.2. SCM as a Structural Dependency Abstraction

To enhance interpretability and logical traceability, we adopt an **SCM-inspired** structural dependency abstraction to represent the reasoning process, drawing inspiration from causal abstractions in neural networks (Geiger et al., 2021). We do not claim to perform full causal identification, intervention, or counterfactual reasoning. Instead, we use the SCM notation to impose an explicit topological organization over evidence, intermediate conclusions, and final verdicts. The model is defined as a tuple $\mathcal{M} = (\mathcal{U}, \mathcal{V}, \mathcal{F})$, where:

- **Exogenous Variables** $\mathcal{U}$: Objective facts retrieved from evidence documents (e.g., text snippets) (Gao et al., 2023). As root nodes, they serve as the non-derived foundational inputs for the reasoning process.

- **Endogenous Variables** $\mathcal{V}$: Intermediate conclusions and the final verdict derived via logical deduction. Each $v_i \in \mathcal{V}$ bridges the gap between evidence and the claim based on its parent variables.

- **Structural Functions** $\mathcal{F}$: A set of mappings $\{f_i\}$ where each function explicitly defines the dependency of an endogenous variable $v_i$ on its parents $Pa(v_i)$, ensuring every step is evidence-grounded.

In this work, we reformulate the MHFV task not merely as a label prediction problem, but as a **constructive structural reasoning process** rooted in the SCM-inspired framework $\mathcal{M}$. Specifically, the verification process initializes with the set of exogenous variables $\mathcal{U}$. It then proceeds recursively: at each reasoning depth, the model applies structural functions $\mathcal{F}$ to existing variables to synthesize new endogenous variables $\mathcal{V}$. This iterative derivation continues until a terminal endogenous variable $y \in \mathcal{V}$, representing the final verification conclusion, is obtained. Consequently, the verdict is derived not as an opaque output, but as the consequence of a clearly defined dependency chain.

To operationalize this framework, the model explicitly constructs and maintains a dynamic **dependency graph** throughout the inference phase. Formally, this graph is a DAG where nodes correspond to variables ($\mathcal{U} \cup \mathcal{V}$) and directed edges denote the dependencies encoded by $\mathcal{F}$. A critical mechanism here is the enforcement of **structural validity constraints**: at any reasoning step $t$, the model is permitted to incorporate a new endogenous variable $v_t$ into the graph *if and only if* its requisite parent set $Pa(v_t)$ is fully present in the current graph structure. This mechanism acts as a logical gatekeeper, ensuring that the reasoning follows a topological order. By doing so, it reduces unsupported logical leaps and makes the generated reasoning path more structurally transparent and easier to inspect.

### 3.3. Data Construction

To endow the model with the capability of reasoning based on the SCM-inspired structure, it is essential to curate a high-quality dataset containing explicit inference chains. We adopt a **distillation-based approach**, where we leverage the superior in-context learning capabilities of a large teacher model to generate training data for the smaller target model. The automated data construction pipeline is illustrated in Figure 3. The process consists of three distinct stages:

**Seed Data Preparation.** We utilize established high-quality multi-hop fact verification benchmarks as our seed

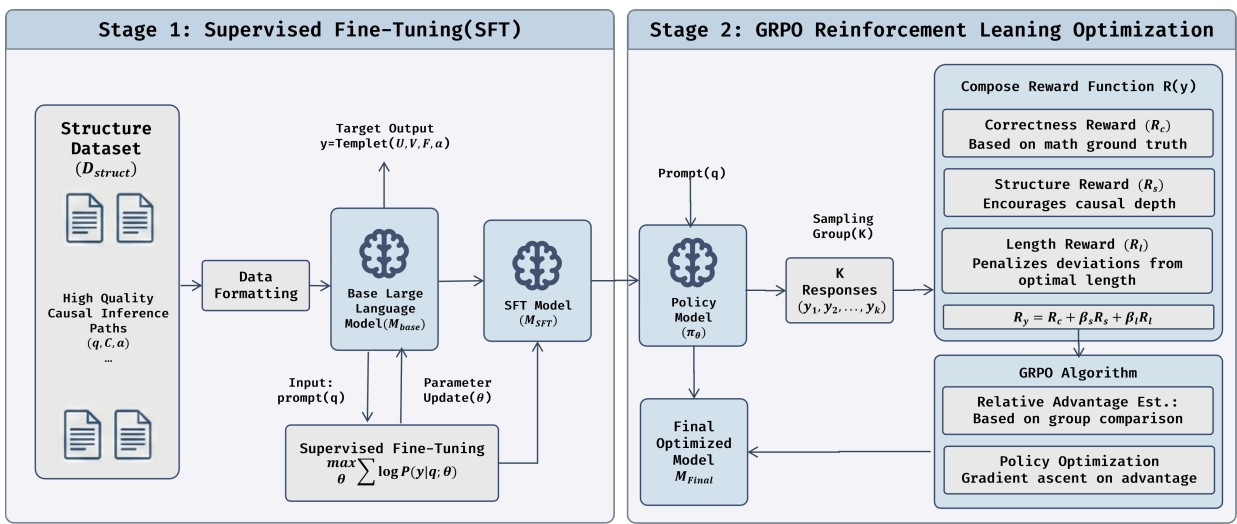

*Figure 2.* The overall architecture of the proposed framework. The training pipeline consists of two stages: (1) **SFT**, where the base model is aligned with the SCM-inspired reasoning paradigm using the structured dataset $\mathcal{D}_{struct}$; and (2) **GRPO reinforcement learning optimization**, where the policy model is refined via group-wise sampling and a composite reward function ($R_c$, $R_s$, $R_l$) to improve reasoning robustness and reduce unsupported structural expansion.

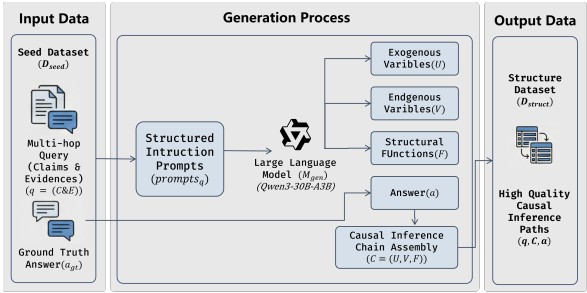

*Figure 3.* The pipeline of automated data construction. The process transforms standard multi-hop queries from the seed dataset ($\mathcal{D}_{seed}$) into structured inference paths. By leveraging structured prompts, a generator LLM explicitly produces SCM-inspired components (Exogenous Variables $\mathcal{U}$, Endogenous Variables $\mathcal{V}$, and Structural Functions $\mathcal{F}$), which are then assembled and filtered to create the high-quality structured dataset ($\mathcal{D}_{struct}$).

dataset, denoted as $\mathcal{D}_{seed}$. Each sample in this dataset comprises a multi-hop query $q$ (encompassing the claim and retrieved evidences) and a ground-truth verification label $a_{gt}$. In our implementation, $\mathcal{D}_{seed}$ consists of 60% of the training queries sampled from HoVer and EX-FEVER. This seed subset provides diverse hop-level reasoning patterns while keeping the distillation cost manageable.

**SCM Component Generation (Distillation).** To extract the SCM-inspired components defined in Section 3, we employ a **structured instruction prompting** strategy. We feed the query $q$ into a powerful teacher LLM, $M_{gen}$ (Qwen3-30B-A3B in our experiments) (Yang et al., 2025). Instead of directly predicting the label, the model is instructed to explicitly generate the set of exogenous variables $\mathcal{U}$, endoge-

nous variables $\mathcal{V}$, and the corresponding structural functions $\mathcal{F}$, culminating in a predicted answer $a$. This step externalizes the latent reasoning process of the teacher model into a structured format (He et al., 2024), which is then used to train the student model.

**Structured Chain Assembly and Validation.** To ensure the reliability of the constructed data, we perform a rigorous assembly and filtering process on the raw outputs:

- **Consistency Filtering:** We first compare the generated answer $a$ with the ground truth $a_{gt}$. Samples where $a \neq a_{gt}$ are discarded to prevent the propagation of erroneous reasoning logic.

- **Sequential Assembly:** For the valid samples, we reorganize the generated components ($\mathcal{U}, \mathcal{V}, \mathcal{F}$) into the sequential decision-making format defined above. Specifically, we serialize the graph construction process into a reasoning chain $\mathcal{C} = \{(\tau_t, \alpha_t, o_t)\}_{t=1}^{N}$, where each step corresponds to the derivation of an endogenous variable via its structural function.

The teacher model generated 10,539 structured candidates in total. We applied ground-truth consistency filtering and removed 96 samples whose predicted labels did not match the gold labels. We observed no formatting errors and only 3 length-related invalid samples, which were also discarded. This filtering step is designed to reduce the propagation of teacher-induced logical noise: a weaker teacher may reduce the yield rate, but inconsistent generated chains are filtered before SFT.

Through this pipeline, we obtain a structure-aware dataset $\mathcal{D}_{struct}$, comprising high-quality inference paths $(q, \mathcal{C}, a)$, which serves as the foundation for the subsequent supervised fine-tuning phase. Prompt templates and serialization examples are provided in Appendix A.

### 3.4. Supervised Fine-Tuning

Upon acquiring the structured training dataset $\mathcal{D}_{struct}$, we initiate the training phase with SFT to align the base LLM, $\mathcal{M}_{base}$, with the SCM-inspired reasoning paradigm. We convert each sample $(q, \mathcal{C}, a)$ in the dataset into a standardized training instance. Specifically, the model input consists of an instruction prompt $prompt(q)$ concatenated with the query $q$, while the target output $y$ is constructed as a structured sequence:

$$y = \text{Template}(\mathcal{U}, \mathcal{V}, \mathcal{F}, a) \tag{1}$$

where $\text{Template}(\cdot)$ denotes a predefined textual template that organizes the set of exogenous variables $\mathcal{U} = \{u_1, \ldots, u_m\}$, the set of endogenous variables $\mathcal{V} = \{v_1, \ldots, v_n\}$, the set of structural functions $\mathcal{F} = \{f_1, \ldots, f_n\}$, and the final answer $a$ into a fixed order and format.

The objective of SFT is to maximize the likelihood probability of the model generating the correct structured sequence given the input. This objective function can be formulated as:

$$\mathcal{L}_{\text{SFT}}(\theta) = -\sum_{(x,y) \in \mathcal{D}_{\text{struct}}} \sum_{t=1}^{|y|} \log P(y_t | x, y_{<t}; \theta) \tag{2}$$

where $\theta$ represents the model parameters, $x$ denotes the input context, and $y_t$ is the token at step $t$. Through training on this objective, the fine-tuned model $\mathcal{M}_{SFT}$ learns to generate traceable reasoning chains that follow the structural logic of "*Identify Evidence* $(\mathcal{U}) \rightarrow$ *Stepwise Derivation* $(\mathcal{V}, \mathcal{F}) \rightarrow$ *Draw Conclusion* $(a)$", thereby establishing a foundation for interpretable verification (Wu et al., 2023).

### 3.5. Optimization via GRPO

To achieve stable and efficient optimization of the model policy during the reinforcement learning phase, we employ the **GRPO** algorithm (Shao et al., 2024; DeepSeek-AI, 2024). By performing group-wise comparisons of multiple outputs generated from the same prompt, GRPO utilizes relative reward signals to update the policy, effectively reducing training variance and enhancing alignment efficiency.

We formalize the reasoning sequence generation as a reinforcement learning task. Given a prompt $q$, the policy model $\pi_\theta$ generates a complete reasoning sequence $y$. The quality of this sequence is evaluated by a composite reward function $R(y)$, comprising three designed components:

**Correctness Reward** ($R_c$). This component assesses the accuracy of the generated answer. Let $a_{pred}$ be the answer parsed from sequence $y$, and $a_{gt}$ be the ground truth. The reward is defined as:

$$R_c(y) = \mathbb{I}(\text{match}(a_{pred}, a_{gt})) \cdot r_{correct} \tag{3}$$

where $\mathbb{I}(\cdot)$ is the indicator function, which equals 1 when the answers match and 0 otherwise; $r_{correct}$ is a fixed high-value positive reward.

**Structure Reward** ($R_s$). To incentivize the model to construct efficient dependency structures, we design a reward based on the principle of Occam's Razor. We encourage the model to rely more on evidence ($\mathcal{U}$) while keeping intermediate reasoning ($\mathcal{V}$) concise. We define the variable quantity difference $\Delta(y) = |\mathcal{U}| - |\mathcal{V}|$. The reward is defined as:

$$R_s(y) = \gamma \cdot \tanh\left(\frac{\Delta(y)}{\delta}\right) \tag{4}$$

where $\gamma$ is the reward coefficient, and $\delta$ is a scaling parameter for normalization. A higher $\Delta(y)$ implies that the model is grounding its conclusion in more direct evidence relative to the number of inferred steps, thus penalizing excessive intermediate reasoning or potential hallucination loops often found in long CoT chains.

**Length Reward** ($R_l$). This reward guides the reasoning chain length $L(y)$ towards an optimal interval $[l_{min}, l_{max}]$ to mitigate the hallucination risks associated with excessively long chains. It is defined as:

$$R_l(y) = -\lambda \cdot \text{dist}(L(y), [l_{min}, l_{max}]) \tag{5}$$

where $\lambda$ is a penalty coefficient, and $\text{dist}(x, [a, b])$ calculates the distance from point $x$ to the interval $[a, b]$ (returns 0 if $x$ is within the interval). In our implementation, $L(y)$ is computed as the generated reasoning length after tokenization. The interval $[l_{min}, l_{max}]$ is estimated from the empirical distribution of valid, ground-truth-matched reasoning chains in $\mathcal{D}_{struct}$. This avoids manually tuning the boundary on the test set and reduces the risk of dataset-specific overfitting.

The final reward is a weighted sum of the three components:

$$R(y) = R_c(y) + \beta_s R_s(y) + \beta_l R_l(y) \tag{6}$$

where $\beta_s$ and $\beta_l$ are hyperparameters balancing the weights.

The core of GRPO lies in estimating policy gradients using the relative performance of samples within a group, thereby reducing the variance associated with single reward signals in traditional methods. Specifically, for a given prompt $q$, we sample $K$ independent reasoning sequences from the current policy $\pi_\theta$ to form a group $G = \{y_1, y_2, \ldots, y_K\}$.

The optimization objective is to maximize the following group relative advantage objective:

$$\mathcal{J}(\theta) = \mathbb{E}_{q \sim \mathcal{D}, G \sim \pi_\theta} \left[ \frac{1}{K} \sum_{i=1}^{K} \frac{\pi_\theta(y_i|q)}{\pi_{\theta_{old}}(y_i|q)} \hat{A}(y_i) \right] \quad (7)$$

where $\hat{A}(y_i)$ is the relative advantage estimate for sample $y_i$, calculated as:

$$\hat{A}(y_i) = \frac{R(y_i) - \text{mean}(\{R(y_j)\}_{j=1}^{K})}{\text{std}(\{R(y_j)\}_{j=1}^{K}) + \epsilon} \quad (8)$$

This advantage estimate reflects the performance of sample $y_i$ relative to the average performance of other samples in the group. By optimizing this objective via gradient ascent, the policy is encouraged to increase the probability of high-reward sequences while suppressing low-reward ones. Additionally, we introduce a KL-divergence term to maintain exploration and prevent premature convergence. Ultimately, GRPO achieves robust and efficient optimization of complex multi-hop reasoning strategies through stable group-wise mechanisms.

## 4. Experiments

### 4.1. Datasets and Benchmarks

To comprehensively evaluate the efficacy of our proposed framework, we conducted experiments on two widely used multi-hop fact verification benchmarks: **HoVer** (Jiang et al., 2020) and **EX-FEVER** (an extension of FEVER (Ma et al., 2024)). HoVer contains 2-hop, 3-hop, and 4-hop reasoning subsets, while EX-FEVER natively supports up to 3-hop reasoning. Therefore, we report EX-FEVER results on 2-hop and 3-hop settings.

### 4.2. Baselines

In our experimental design, we compare our approach against a diverse set of representative baseline methods, covering various reasoning paradigms:

- **Direct Prediction (DP)**: An end-to-end method that directly predicts the verdict based on retrieved evidence without explicit reasoning steps (Achiam et al., 2023).

- **ProgramFC** (Pan et al., 2023): A programmatic approach that utilizes LLMs to generate executable scripts, enforcing structured multi-step reasoning.

- **FOLK**: A formal reasoning framework that grounds the verification process in First-Order Logic (FOL) rules.

- **RAG**: The standard Retrieval-Augmented Generation paradigm (Lewis et al., 2020), including recent variants

like Self-RAG (Asai et al., 2024) and CRAG (Yan et al., 2024).

- **Decompose-Verify**: A strategy that decomposes complex composite claims into simpler sub-claims for independent verification (Zhou et al., 2023).

- **QACheck**: An interrogation-based method that guides the reasoning process by generating and answering intermediate questions.

- **Search-o1**: A strong baseline combining iterative retrieval with the zero-shot reasoning capabilities of large language models (Li et al., 2025b).

To ensure a fair comparison, we additionally reproduce DP and ProgramFC using the same **Qwen3-8B** backbone as our method. For external baselines originally evaluated with GPT-3.5, we report their results under their original settings and separate them from same-backbone comparisons.

### 4.3. Main Results

Table 1 presents the comparative performance of our proposed framework against various baselines on the HoVer and EX-FEVER benchmarks. Overall, **SCM-GRPO** achieves the best average accuracy and consistently ranks first across all five evaluation subsets.

**Superiority over Baselines.** SCM-GRPO achieves the best average accuracy and consistently ranks first across all evaluation subsets. The gains are especially meaningful because the model preserves an explicit dependency-graph reasoning trace, whereas several external baselines rely on proprietary GPT-3.5 settings and do not provide the same structural traceability. Under the same Qwen3-8B backbone, SCM-GRPO substantially improves over both DP and ProgramFC, supporting that the gains are not merely caused by backbone differences.

**Validation of RL Optimization.** A critical observation from the "Pure LLM" section corroborates the central hypothesis of this paper. Simply incorporating the SCM-CoT mechanism *without* RL optimization often leads to performance deterioration. For example, applying SCM-CoT to Qwen3-14B causes a sharp drop in accuracy from 57.48% to 47.22% on the HoVer 2-hop task. This empirical evidence confirms that unconstrained structural complexity can introduce noise and reduce performance. In contrast, the complete **SCM-GRPO** framework mitigates this issue through multi-dimensional reward optimization, converting structural traceability into tangible performance gains.

**Unified Ablation Analysis.** Table 2 summarizes both the optimization-strategy ablation and the reward-component

*Table 1.* Main verification accuracy (%) on HoVer and EX-FEVER benchmarks. Same-backbone baselines are evaluated using Qwen3-8B. External baselines are reported under their original GPT-3.5 settings.

| Category | Methods | HoVer | | | EX-FEVER | | Avg. |
|---|---|---|---|---|---|---|---|
| | | 2-hop | 3-hop | 4-hop | 2-hop | 3-hop | |
| Pure LLM | Qwen3-8B | 56.29 | 57.60 | 48.28 | 65.85 | 64.01 | 58.41 |
| | Qwen3-8B (SCM-CoT) | 56.43 | 47.20 | 48.36 | 59.73 | 58.38 | 54.02 |
| | Qwen3-8B (ProgramFC) | 64.76 | 62.85 | 59.45 | 68.42 | 66.08 | 64.31 |
| | Qwen3-14B | 57.48 | 54.88 | 48.31 | 69.40 | 62.79 | 58.57 |
| | Qwen3-14B (SCM-CoT) | 47.22 | 53.49 | 48.12 | 57.16 | 61.08 | 53.41 |
| | Llama3.2-3B | 46.18 | 52.75 | 47.35 | 63.95 | 56.03 | 53.25 |
| | Llama3.2-3B (SCM-CoT) | 44.67 | 50.90 | 47.64 | 47.04 | 49.94 | 48.04 |
| External Baselines (GPT-3.5) | Direct Prediction (DP) | 72.56 | 61.70 | 59.57 | 81.03 | 73.02 | 69.58 |
| | ProgramFC | 66.84 | 55.35 | 52.60 | 71.60 | 62.40 | 61.76 |
| | FOLK | 67.60 | 61.20 | 55.20 | 75.80 | 68.40 | 65.64 |
| | RAG | 59.20 | 56.60 | 55.20 | 69.00 | 64.80 | 60.96 |
| | Decompose-Verify | 62.60 | 57.31 | 55.60 | 68.40 | 63.00 | 61.38 |
| | QACheck | 67.60 | 60.60 | 59.00 | 75.60 | 68.60 | 66.28 |
| | Search-o1 | 69.00 | 59.80 | 56.60 | 77.80 | 72.80 | 67.20 |
| **Ours** | **SCM-GRPO** | **73.42** | **63.15** | **60.88** | **82.66** | **75.00** | **71.02** |

ablation. In the upper block, the SFT baseline yields an accuracy of 73.33%, while CPO degrades performance to 71.91%, suggesting that pairwise preference optimization struggles to capture subtle logical differences in multi-hop reasoning chains. In contrast, SCM-GRPO achieves the highest accuracy among the optimization strategies, indicating that group-wise relative optimization is more effective for stabilizing structured reasoning. In the lower block, removing the structure reward $R_s$ reduces accuracy from 0.7544 to 0.7427, showing that explicit structural regularization contributes beyond correctness and length alone. Removing the length reward $R_l$ causes a much larger degradation to 0.6433 and leads to reward saturation and gradient vanishing, indicating that $R_l$ is critical for preventing uncontrolled reasoning length and maintaining stable optimization.

**Efficiency and Simpler Alternatives.** We also considered simpler alternatives such as decoding-time length penalties and rule-based reranking. A decoding-time length penalty is coarse because it penalizes all long outputs, including valid long reasoning chains required by difficult instances. Rule-based reranking can select structurally compact candidates, but it requires sampling multiple candidates at inference time, increasing inference cost. In contrast, SCM-GRPO distills the structural and length preferences into model parameters during training. Thus, at inference time, the model retains a single-pass generation procedure while producing more compact and traceable reasoning chains.

## 5. Conclusion

To address the challenges of opaque logical reasoning mechanisms and susceptibility to hallucinations in complex multi-hop fact verification, this paper proposes a framework integrating an SCM-inspired structural dependency graph with RL optimization. We first formalize the multi-hop verification process as an explicit reasoning task over evidence variables, intermediate conclusions, and structural functions. By constructing high-quality supervised fine-tuning data through a filtered distillation pipeline from a teacher model, we empower the student model to generate structured and traceable reasoning chains. To mitigate the risks of reasoning-chain redundancy and unsupported intermediate steps, we further design a rule-based RL optimization strategy. Incorporating the GRPO algorithm, this strategy dynamically guides the model to strike an optimal balance among correctness, structural rationality, and length compliance.

Experimental results demonstrate that our proposed method achieves state-of-the-art or highly competitive performance across multiple multi-hop fact verification benchmarks, including HoVer and EX-FEVER. Ablation studies substantiate that the GRPO-based optimization strategy effectively enhances reasoning accuracy and robustness. Furthermore, our in-depth analysis of reasoning-chain length and structural complexity provides novel empirical insights for understanding and improving the reasoning behaviors of LLMs.

*Table 2.* Unified ablation study on the Qwen3-8B backbone. The upper block compares optimization strategies, while the lower block analyzes the contribution of reward components.

| Category | Setting | Accuracy | Training Status |
|---|---|---|---|
| Optimization Strategy | Qwen3-8B-SFT | 0.7333 | Stable |
| | Qwen3-8B-SFT + CPO | 0.7191 | Degraded |
| | Qwen3-8B-SFT + GRPO (Ours) | **0.7542** | Stable |
| Reward Component | Full reward ($R_c + \beta_s R_s + \beta_l R_l$) | **0.7544** | Stable |
| | w/o $R_s$ ($\beta_s = 0$) | 0.7427 | Stable |
| | w/o $R_l$ ($\beta_l = 0$) | 0.6433 | Reward saturation / gradient vanishing |
| | $\beta_s = 1.0$ | 0.6959 | Unstable structural penalty |
| | $\beta_l = 0.5$ | 0.6901 | Reward collapse / length explosion |

The primary limitation of this work lies in its partial dependence on the quality of automatically constructed structured training data. Additionally, the generalization capability for ultra-long reasoning chains (e.g., exceeding 4 hops) remains to be further verified. Future work will explore more robust data construction methodologies and extend this structural reasoning framework to a broader spectrum of complex reasoning tasks.

## Acknowledgements

This work was supported by the National Key Research and Development Program of China under Grant No. 2024YFC3308101, as part of the project "Long- and Short-Term Holographic Profiling of Bond Investors Based on Trading Behavior Characteristics," with support from Xinjiang Future Enterprise Incubator Co., Ltd. The authors thank the anonymous reviewers for their constructive comments.

## Impact Statement

This paper aims to improve the reliability and traceability of multi-hop fact verification systems. By encouraging models to ground intermediate reasoning steps in explicit evidence-dependent structures, the proposed framework may help reduce unsupported explanations and improve the auditability of automated fact-checking tools. At the same time, fact verification systems can still reflect biases or omissions in the underlying evidence sources, datasets, and teacher-generated training data. Therefore, the proposed method should be used as an assistive tool rather than as a replacement for human judgment in high-stakes information verification scenarios.

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

## A. Prompt Template and Serialization Example

**Teacher Prompt.** Given a claim and retrieved evidence, identify: (1) exogenous variables $\mathcal{U}$ directly supported by evidence; (2) endogenous variables $\mathcal{V}$ derived from existing variables; (3) structural functions $\mathcal{F}$ specifying the parent variables used to derive each endogenous variable; and (4) the final verification label. The teacher model is instructed not to introduce facts unsupported by the retrieved evidence.

**Output Format.**

```
Exogenous Variables:
U1: <evidence-grounded fact>
U2: <evidence-grounded fact>

Endogenous Variables:
V1: <intermediate conclusion derived from existing variables>
V2: <intermediate conclusion derived from existing variables>

Structural Functions:
f1: V1 <- {U1, U2}
f2: V2 <- {V1, U3}

Final Answer:
Supported / Refuted
```

**Serialization.** After filtering, each valid graph is serialized into the SFT target sequence in topological order. This ensures that every endogenous variable appears only after its declared parent variables have already been introduced. The resulting sequence follows the pattern: evidence identification → intermediate derivation → final answer.

## B. Training Stability and Reward Sensitivity

*Table 3.* Training stability statistics of SCM-GRPO.

| Metric | Value |
|---|---|
| RL steps | 1700 |
| Reward last std. | 0.259 |
| Final loss | 0.00018 |
| Final gradient norm | 0.191 |

The results in Table 4 show that the composite reward is sensitive to nonlinear coupling among correctness, structural regularization, and length control. The best performance appears only under the joint configuration, validating the need for multi-objective reward design. Excessively large length penalties collapse reward dynamics, while removing length control leads to uncontrolled reasoning expansion and unstable optimization.

## C. Structural Faithfulness Evaluation

**Structural Faithfulness Rate.** We further evaluate whether each generated endogenous node is grounded in its declared parent evidence or intermediate variables. A reasoning chain is counted as structurally faithful if all generated structural functions satisfy the DAG validity constraint and every endogenous variable has its required parent nodes present before generation. SCM-GRPO achieves a structural faithfulness rate above 95%, indicating that the improvement is not merely due to better final-label matching but also to more valid intermediate reasoning structures.

*Table 4.* Reward and accuracy under different hyperparameter settings.

| Series | Name | Accuracy | Status |
|---|---|---|---|
| $r_{correct}$ | $r_{correct} = 0$ | 0.7018 | Normal |
| | $r_{correct} = 5$ | 0.7018 | Normal |
| | $r_{correct} = 10$ | 0.7485 | Normal |
| | $r_{correct} = 20$ | **0.7544** | Normal (Best) |
| | $r_{correct} = 40$ | 0.7485 | Normal |
| | $r_{correct} = 80$ | 0.7018 | Normal |
| $\beta_s$ | $\beta_s = 0.0$ | 0.7427 | Normal |
| | $\beta_s = 0.25$ | 0.7427 | Normal |
| | $\beta_s = 0.5$ | **0.7544** | Normal (Best) |
| | $\beta_s = 1.0$ | 0.6959 | Unstable structural penalty |
| $\beta_l$ | $\beta_l = 0$ | 0.6433 | Reward saturated, gradient vanishing |
| | $\beta_l = 0.1$ | 0.6901 | Normal |
| | $\beta_l = 0.2$ | **0.7544** | Normal (Best) |
| | $\beta_l = 0.5$ | 0.6901 | Reward collapsed, length exploded |
| | $\beta_l = 1.0$ | 0.6959 | Large reward fluctuation |
| | $\beta_l = 1.5$ | 0.6959 | Reward severely unstable |
| | $\beta_l = 2.0$ | 0.6959 | Reward unstable |
| Length range | 80–160 tokens | **0.7544** | Normal (Best) |
| | 120–240 tokens | 0.6901 | Normal |
| | 160–320 tokens | 0.6959 | Normal |

# D. Empirical Analysis of Structural Reasoning

In this section, we provide a detailed statistical analysis of the reasoning structures generated by the SFT baseline and our proposed SCM-GRPO framework. We focus on the distribution of variable types, the complexity of dependency paths, and their correlation with verification accuracy.

## D.1. Impact of Optimization on Structural Complexity

We first examine the aggregate structural characteristics and their resulting performance. As shown in Figure 4a, SCM-GRPO demonstrates a distinct shift in reasoning topology compared to SFT. SCM-GRPO explicitly generates a higher number of **Exogenous Variables** (Evidence nodes) while significantly reducing the number of **Endogenous Variables** (Intermediate inference nodes) and total **Dependency Paths**. This indicates that GRPO optimization encourages the model to ground its reasoning more heavily in direct evidence rather than constructing long, potentially unsupported inference chains.

Despite this reduction in structural complexity, Figure 4b confirms that model performance is not compromised. SCM-GRPO achieves a verification accuracy of **70.35%**, slightly outperforming the SFT baseline (69.16%). Figure 4c further illustrates distribution stability, where SCM-GRPO eliminates the long-tail outliers observed in SFT.

## D.2. Relationship Between Structural Expansion and Topology

To understand the mechanism behind the accuracy maintenance shown in Figure 4b despite structural pruning shown in Figure 4c, we analyze the correlation between variable generation and path formation.

Figure 5 reveals a fundamental divergence in reasoning mechanisms. The SFT model shows a strong positive linear correlation between the total number of variables and the number of dependency paths. This implies that for every new piece of information the SFT model processes, it tends to increase the complexity of its logical dependencies. Conversely,

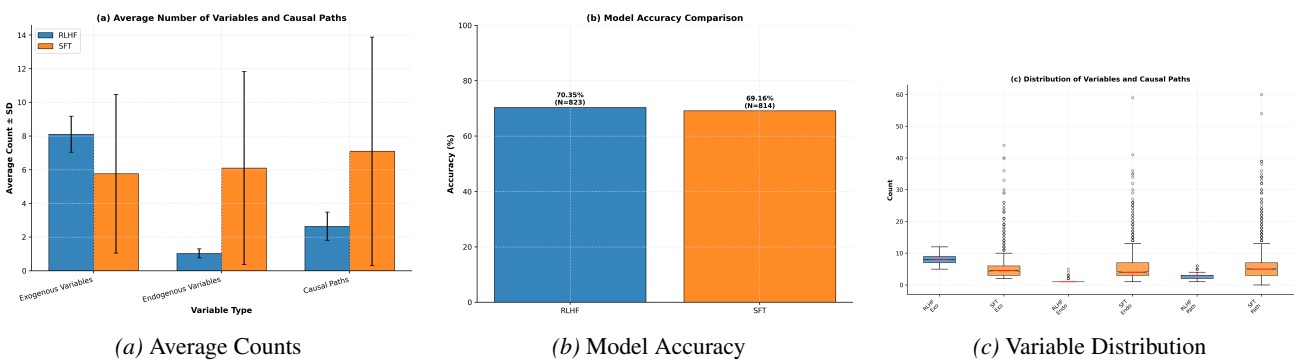

*(a)* Average Counts   *(b)* Model Accuracy   *(c)* Variable Distribution

*Figure 4.* **Structural Complexity and Performance Overview.** (a) SCM-GRPO significantly reduces endogenous variables and paths compared to SFT. (b) Despite structural pruning, SCM-GRPO maintains superior accuracy (70.35% vs 69.16%). (c) SCM-GRPO exhibits a stable, compact distribution, whereas SFT shows numerous high-complexity outliers.

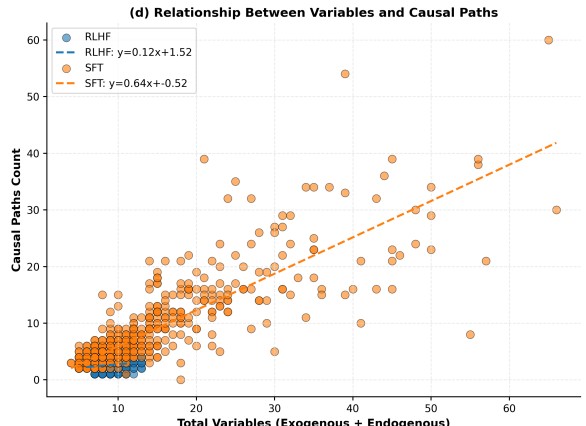

*Figure 5.* **Relationship Between Variables and Dependency Paths.** The sharp slope for SFT ($y = 0.64x$) indicates complexity expansion, while the flat slope for SCM-GRPO ($y = 0.12x$) suggests efficient evidence integration.

SCM-GRPO exhibits a much flatter slope. This "decoupling" effect indicates that our method can incorporate more evidence (exogenous variables) without proportionally inflating the complexity of the reasoning graph.

### D.3. Statistical Significance and Internal Correlations

Finally, we verify the statistical validity of these structural shifts. Figure 6a confirms that the observed differences in Exogenous Variables, Endogenous Variables, and Dependency Paths between SFT and SCM-GRPO are statistically highly significant ($p < 1e - 30$). This suggests that the changes in reasoning behavior are a systematic result of GRPO alignment.

Figure 6b provides a heatmap of Pearson correlation coefficients within each model. The SFT model exhibits a high correlation coefficient of **0.85**, reinforcing that its reasoning structure is tightly coupled and prone to complexity expansion. In contrast, SCM-GRPO shows a lower correlation of **0.16**, supporting the conclusion that SCM-GRPO successfully learns to prioritize evidence grounding over unnecessary logical elaboration.

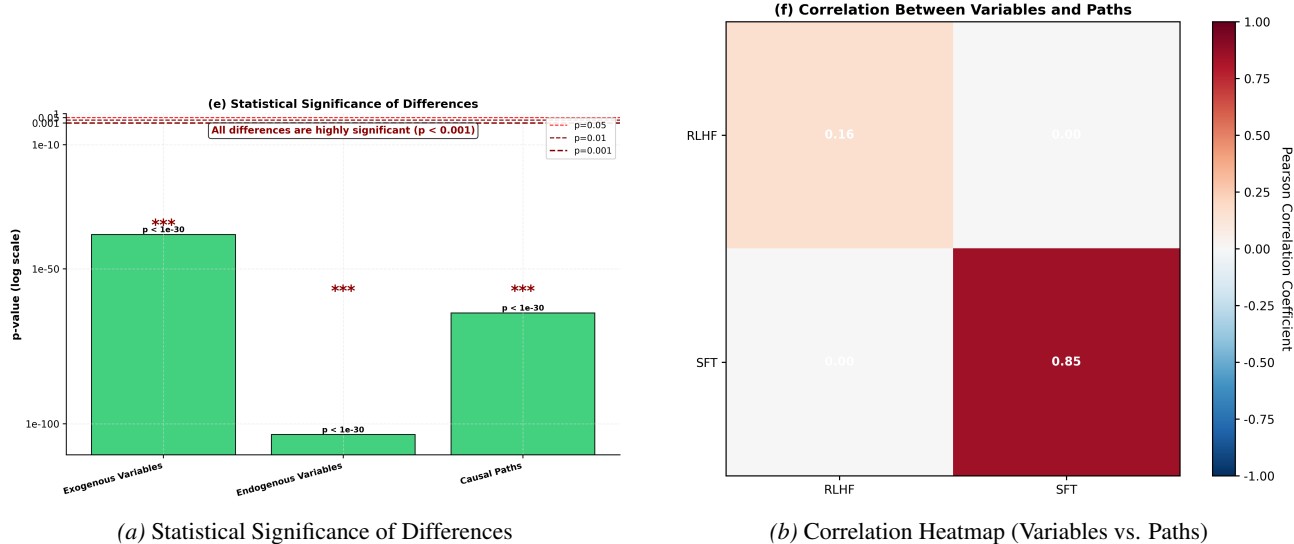

*(a)* Statistical Significance of Differences

*(b)* Correlation Heatmap (Variables vs. Paths)

*Figure 6.* **Statistical Validation.** (e) Differences in structural components are statistically significant ($p < 1e - 30$). (f) SFT shows high internal correlation (0.85) between variables and paths, while SCM-GRPO (0.16) effectively decouples them.

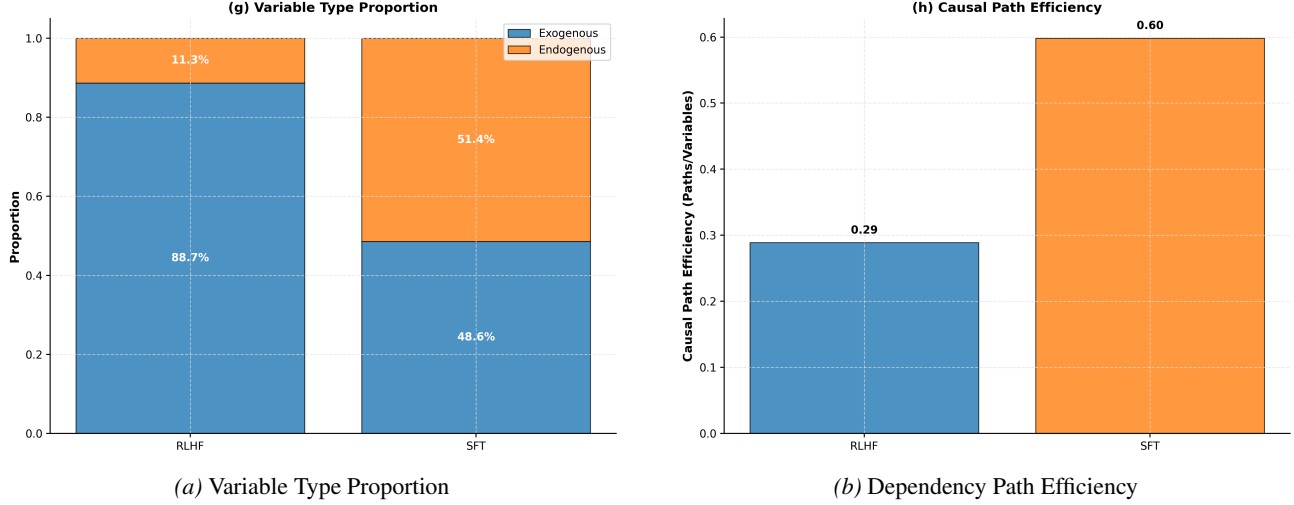

*(a)* Variable Type Proportion

*(b)* Dependency Path Efficiency

*Figure 7.* **Structural Efficiency Metrics.** (g) SCM-GRPO shifts the focus to exogenous evidence (88.7%). (h) SCM-GRPO improves path efficiency (0.29 paths/var) compared to SFT (0.60 paths/var).

