# A. Empirical Analysis of Structural Causal Reasoning

In this section, we provide a detailed statistical analysis of the reasoning structures generated by the SFT baseline and our proposed SCM-GRPO framework (denoted as **RLHF** in the figures). We focus on the distribution of variable types, the complexity of causal paths, and their correlation with verification accuracy.

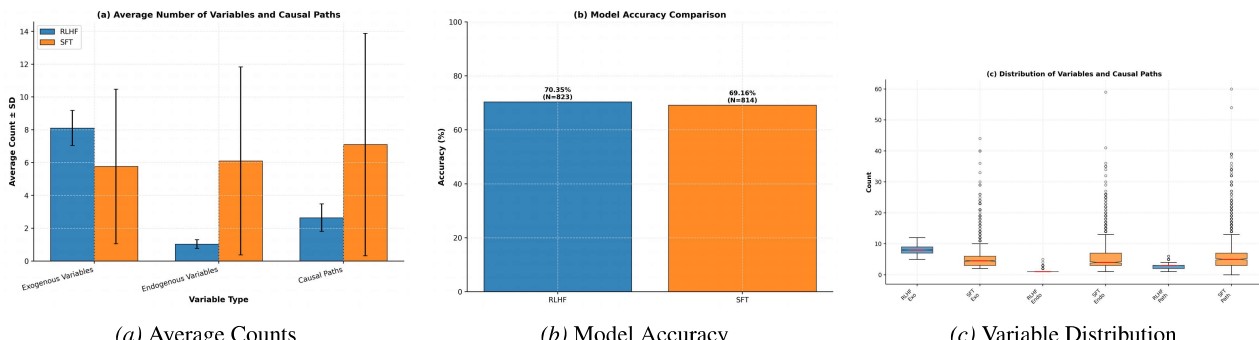

*(a)* Average Counts      *(b)* Model Accuracy      *(c)* Variable Distribution

*Figure 4.* **Structural Complexity and Performance Overview.** (a) RLHF significantly reduces endogenous variables and paths compared to SFT. (b) Despite structural pruning, RLHF maintains superior accuracy (70.35% vs 69.16%). (c) RLHF exhibits a stable, compact distribution, whereas SFT shows numerous high-complexity outliers.

## A.1. Impact of Optimization on Structural Complexity (Figs. 4a-4c)

We first examine the aggregate structural characteristics and their resulting performance. As shown in Figure 4a, the RLHF model demonstrates a distinct shift in reasoning topology compared to SFT. RLHF explicitly generates a higher number of **Exogenous Variables** (Evidence nodes) while significantly reducing the number of **Endogenous Variables** (Intermediate inference nodes) and total **Causal Paths**. This indicates that the GRPO optimization encourages the model to ground its reasoning more heavily in direct evidence rather than constructing long, potentially hallucinatory inference chains.

Despite this reduction in structural complexity, Figure 4b confirms that the model performance is not compromised. The RLHF model achieves a verification accuracy of **70.35%**, slightly outperforming the SFT baseline (69.16%). Figure 4c further illustrates the distribution stability, where RLHF eliminates the long-tail outliers observed in SFT.

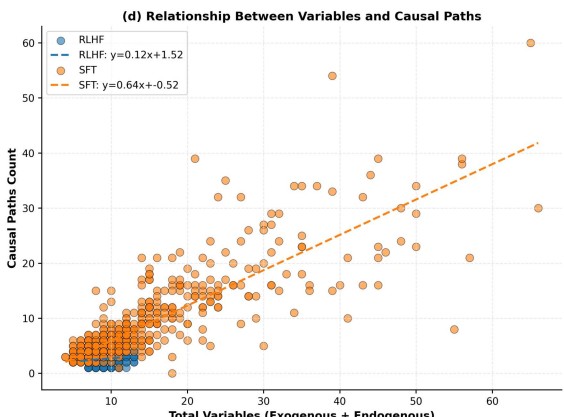

*Figure 5.* **Relationship Between Variables and Causal Paths.** The sharp slope for SFT ($y = 0.64x$) indicates complexity explosion, while the flat slope for RLHF ($y = 0.12x$) suggests efficient evidence integration.

## A.2. Relationship Between Structural Expansion and Topology (Fig. 5)

To understand the mechanism behind the accuracy maintenance shown in Figure 4b despite the structural pruning shown in Figure 4c, we analyze the correlation between variable generation and path formation.

Figure 5 reveals a fundamental divergence in reasoning mechanisms. The SFT model shows a strong positive linear correlation between the total number of variables and the number of causal paths. This implies that for every new piece of information the SFT model processes, it tends to exponentially increase the complexity of its logical dependencies. Conversely, the RLHF model exhibits a much flatter slope. This "decoupling" effect indicates that our method can incorporate more evidence (Exogenous variables) without proportionally inflating the complexity of the reasoning graph.

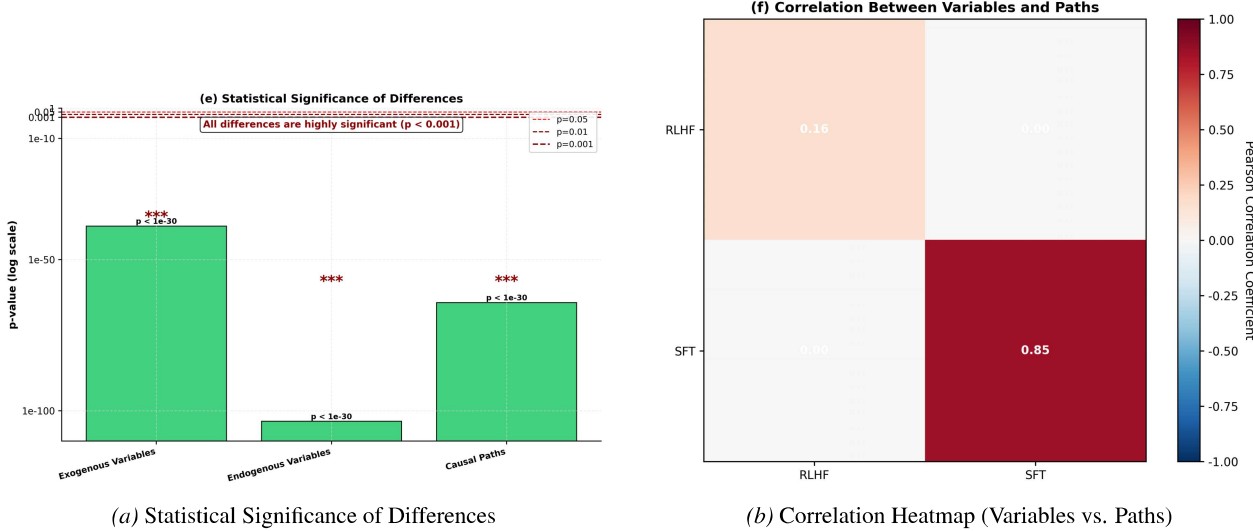

*(a)* Statistical Significance of Differences        *(b)* Correlation Heatmap (Variables vs. Paths)

*Figure 6.* **Statistical Validation.** (e) Differences in structural components are statistically significant ($p < 1e - 30$). (f) SFT shows high internal correlation (0.85) between variables and paths, while RLHF (0.16) effectively decouples them.

**A.3. Statistical Significance and Internal Correlations (Figs. 6a-6b)**

Finally, we verify the statistical validity of these structural shifts. Figure 6a confirms that the observed differences in Exogenous Variables, Endogenous Variables, and Causal Paths between SFT and RLHF are statistically highly significant ($p < 1e - 30$). This proves that the changes in reasoning behavior are a systematic result of the GRPO alignment.

Figure 6b provides a heatmap of Pearson correlation coefficients within each model. The SFT model exhibits a high correlation coefficient of **0.85**, reinforcing that its reasoning structure is tightly coupled and prone to complexity explosion. In contrast, the RLHF model shows a negligible correlation of **0.16**, supporting the conclusion that SCM-GRPO successfully learns to prioritize evidence gathering over unnecessary logical elaboration.

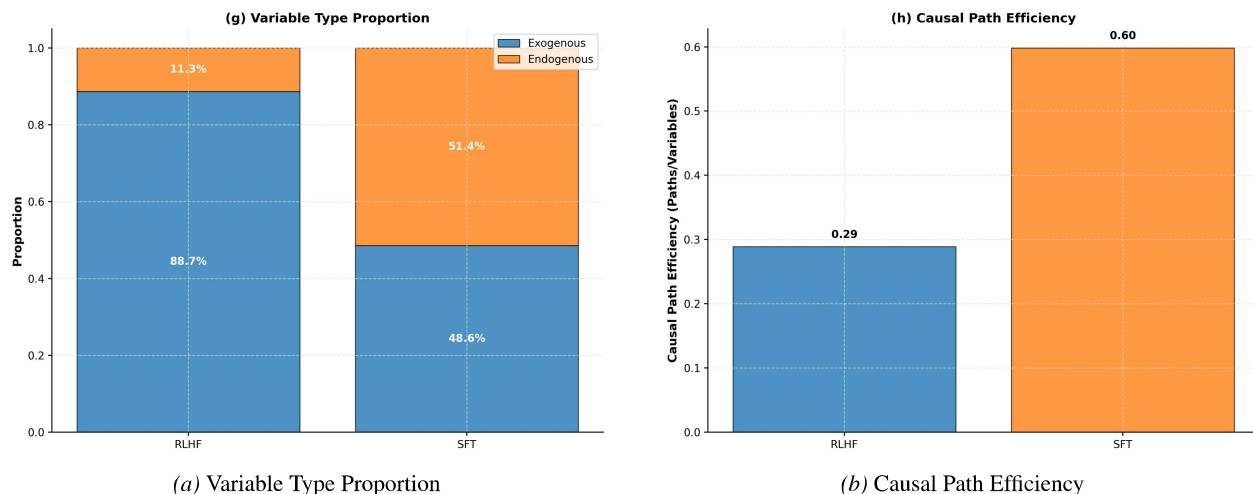

*(a)* Variable Type Proportion                                      *(b)* Causal Path Efficiency

*Figure 7.* **Structural Efficiency Metrics.** (g) RLHF shifts the focus to exogenous evidence (88.7%). (h) RLHF improves path efficiency (0.29 paths/var) significantly compared to SFT (0.60 paths/var).