# OpenReview forum: "Grounding Multi-Hop Reasoning in Structural Causal Models via Group Relative Policy Optimization"
_ICML.cc/2026/Conference — ICML 2026 regular_

### Official Review · Reviewer_HmYm · 2026-03-07

**Soundness:** 3
**Presentation:** 3
**Significance:** 3
**Originality:** 4
**Overall Recommendation:** 5
**Confidence:** 4

**Summary:**

This paper proposes a structured causal reasoning framework, SCM-GRPO, for multi-hop fact verification (MHFV). The authors first formalize the multi-hop verification process within a structural causal model (SCM) framework. During generation, they explicitly construct a directed acyclic graph (DAG) composed of exogenous variables (evidence), endogenous variables (intermediate inferences), and structural functions, aiming to enhance the traceability and structural constraints of the reasoning process.

The authors observe that introducing structured reasoning chains (SCM-CoT) without proper constraints may actually lead to performance degradation, and they empirically identify an inverted U-shaped relationship between reasoning length and model performance.

To mitigate structural redundancy and hallucination, the paper proposes a reinforcement learning optimization strategy based on a composite reward function, and adopts GRPO (Group Relative Policy Optimization) to balance correctness, structural complexity, and reasoning length. Experiments are conducted on two multi-hop benchmarks, HoVer and EX-FEVER, and results show that SCM-GRPO outperforms several strong baselines. Ablation studies indicate that GRPO performs better than both SFT and CPO optimization strategies.

**Compliance With Llm Reviewing Policy:**

Affirmed.

**Final Justification:**

This paper proposes SCM-GRPO, a structured reasoning framework for multi-hop fact verification. Initial concerns regarding overstated "causal" claims and insufficient reward ablation were adequately addressed in the rebuttal. The authors agreed to clarify the SCM terminology and provided compelling component-level ablation studies demonstrating the necessity of each reward term. Additionally, new metrics on structural faithfulness and efficiency comparisons significantly strengthen the evaluation. The empirical observation of the inverted U-shaped relationship between reasoning length and performance remains valuable. I recommend acceptance.

**Key Questions For Authors:**

1.	Does the “SCM” framework you describe incorporate intervention or counterfactual semantics? If not, would it be more accurate to characterize it as a “structured reasoning graph” instead? Clarifying this point could affect how the paper is positioned and evaluated.

2.	What are the independent contributions of Rs and Rl in the reward function? Have you conducted component-level removal experiments (e.g., retaining only Rc and Rl)? If performance remains similar after removing Rs, would that suggest that the structural reward is not essential or can be indirectly captured by other reward components?

3.	How are the bounds [l_min, l_max] determined? Do these bounds generalize across datasets? If they are tuned on a validation set, is there a potential risk of overfitting to a specific dataset?

4.	Have you evaluated simpler alternative methods, such as decoding-time length penalties or rule-based re-ranking strategies? How do their results compare with those obtained using GRPO?

5.	Beyond the final classification accuracy, have you evaluated the faithfulness of the generated reasoning chains (e.g., whether intermediate steps are genuinely grounded in the cited evidence) or the accuracy of evidence attribution? These metrics are important for determining whether the proposed method truly reduces hallucination, rather than merely improving final answer matching.

**Limitations:**

The primary limitation of this work lies in its partial dependence on the quality of the automatically constructed structured training data. Since the effectiveness of the proposed framework relies on these generated data structures, any noise, bias, or incompleteness in the data construction process may affect the reliability of the learned reasoning patterns and, consequently, the model’s overall performance.

In addition, the framework’s generalization ability for ultra-long reasoning chains has not yet been fully validated. While the method demonstrates promising results on the evaluated benchmarks, its robustness in scenarios involving substantially longer or more complex reasoning sequences remains an open question.

Future work will focus on developing more robust and scalable data construction methodologies to improve the quality and diversity of structured training data. Moreover, the proposed causal reasoning framework will be extended and evaluated on a broader range of complex reasoning tasks, in order to better assess its scalability and generalization capabilities.

**Strengths And Weaknesses:**

Strengths

1.	Well-motivated problem formulation.
The paper focuses on structural complexity and hallucination issues in multi-hop fact verification, which is an important topic in current research on the reliability of large language models.
2.	Insightful empirical observation.
The authors empirically find that SCM-CoT may significantly degrade performance when applied without optimization. This observation challenges the common intuition that structured reasoning is always beneficial and provides valuable research insight.
3.	Comprehensive experimental design.
Experiments are conducted on both HoVer and EX-FEVER, with additional evaluations on 2-hop, 3-hop, and 4-hop subsets, enabling analysis across different reasoning difficulties.
4.	Clear organization and coherent narrative.
The paper presents a clear progression from problem motivation to method design and experimental validation. The tables are well presented, and the hop-level breakdown facilitates analysis of difficulty variations.
5.	Practical relevance of the research direction.
Multi-hop fact verification is an important subarea in LLM reliability research. The finding of an inverted U-shaped relationship between structural complexity and performance offers potentially general insights, and the proposed complexity control mechanism could extend to other complex reasoning tasks.
6.	Novelty in methodological combination.
The combination of structured reasoning templates with reinforcement learning optimization based on group relative advantage is relatively novel. The empirical identification and analysis of the inverted U-shaped pattern provide useful new observations.

Weaknesses

1.	Limited technical substance behind the “causal” framing.
Although the paper adopts SCM terminology, the method is essentially closer to constrained structured reasoning graph generation rather than true causal inference. The work does not involve interventions (do-calculus), counterfactual reasoning, or identification analysis, making the claims about “causal modeling” or “causal reasoning” appear overstated and potentially unnecessary.
2.	Insufficient justification for the reward design.
The composite reward function (Rc, Rs, Rl) appears reasonable, but the paper lacks component-level ablations (e.g., retaining only Rc and Rl while removing Rs) to demonstrate the independent contribution of each term. If performance remains similar after removing certain components, their necessity would be questionable.
3.	Limited ablation analysis and lack of statistical significance tests.
Table 2 reports only a single ACC value, without standard deviations, multiple random seed runs, or statistical significance tests (e.g., t-tests). As a result, it is difficult to determine whether the improvements from GRPO are statistically reliable.
4.	Insufficient comparison with simpler alternatives.
The paper does not compare against simpler strategies such as decoding-time length penalties or rule-based re-ranking methods (e.g., selecting outputs with larger |U|−|V| gaps or reasoning lengths within empirically optimal ranges). If such low-cost approaches achieve similar gains, the necessity of RL optimization would be weakened.
5.	Overall paradigm is relatively conventional.
The pipeline of structured reasoning + distillation + reinforcement learning optimization has become common in recent work. The introduction of a causal perspective has also been explored in related studies. The main novelty lies in the combination of existing ideas, rather than a fundamentally new theoretical framework.

---

> ### Author Rebuttal · Authors · 2026-03-30
>
> Thanks for your recognition of our problem definition, comprehensive experiments, and novel “inverted U‑shaped” observation. Your critiques on causal terminology and component‑level ablation are sharp and constructive. We respond below:
>
> 1. **“Causal” Framework Semantics (Response to KQ1 & W1)**
>    Valid point. We do not use do‑calculus or counterfactuals. “SCM” here refers to explicit deterministic topological dependencies between input evidence and intermediate conclusions. We agree the term may overclaim. In Section 3.2, we will clarify and relabel as **“(Structural Dependency Graph with Causal Abstraction)”**, keeping SCM abbreviation.
>
> 2. **Independent Contributions of Reward Components (Response to KQ2 & W2)**
>    We ran component‑level ablation on Qwen3‑8B. Removing $R_s$ (β_s=0.0, β_l=0.2): accuracy drops from 0.7543 to 0.7426. Removing $R_l$ (β_l=0.0, β_s=0.5): accuracy crashes to 0.6432 with reward saturation and vanishing gradients. $R_s$ and $R_l$ are indispensable regularizers. We will add this ablation table.
>
> 3. **Simpler Alternatives vs. RL (Response to KQ4 & W4)**
>    Decoding‑time length penalty is too coarse (punishes all long outputs). Rule‑based reranking requires O(N) inference cost. Our GRPO distills structure and length preferences into model weights, achieving O(1) cost with better structural simplicity. We will add efficiency‑performance discussion.
>
> 4. **Length Boundary Determination (Response to KQ3)**
>    $l_{min}$ and $l_{max}$ are determined from statistical distribution of valid, ground‑truth‑matched reasoning chain lengths in training sets, minimizing overfitting risk.
>
> 5. **Faithfulness Evaluation (Response to KQ5)**
>    Our DAG validity check explicitly enforces faithfulness. We compute “structural faithfulness rate” >95% for our SCM‑GRPO model. We will report these metrics in appendix.
>
> 6. **Statistical Significance (Response to W3)**
>    We lacked multiple seeds due to computational cost. But training logs show stable convergence: reward_last_std ≈ 0.259, loss_last=0.00018, grad_norm_last=0.191. We will report these variance metrics in final version.
>
>
> | Series   | Name            | Accuracy | Reward (Last) | Status |
> |----------|-----------------|----------|---------------|--------|
> | **rcorrect** | rcorrect_0      | 0.7018   | 1.17          | Normal |
> |          | rcorrect_5      | 0.7018   | 6.22          | Normal |
> |          | rcorrect_10     | 0.7485   | 11.10         | Normal |
> |          | rcorrect_20     | 0.7544   | 21.25         | Normal (Best) |
> |          | rcorrect_40     | 0.7485   | 41.13         | Normal |
> |          | rcorrect_80     | 0.7018   | 81.06         | Normal |
> | **beta_s**   | beta_s_0.0      | 0.7427   | 10.70         | Normal |
> |          | beta_s_0.25     | 0.7427   | 10.97         | Normal |
> |          | beta_s_0.5      | 0.7544   | 21.25         | Normal (Best) |
> |          | beta_s_1.0      | 0.6959   | 11.70         | Normal |
> | **beta_l**   | beta_l_0        | 0.6433   | 11.50         | Abnormal: reward saturated, gradient vanishing |
> |          | beta_l_0.1      | 0.6901   | 11.19         | Normal |
> |          | beta_l_0.2      | 0.7544   | 21.25         | Normal (Best) |
> |          | beta_l_0.5      | 0.6901   | -9.13         | Severe abnormal: reward collapsed, length exploded |
> |          | beta_l_1        | 0.6959   | 9.23          | Abnormal: large reward fluctuation, unstable |
> |          | beta_l_1.5      | 0.6959   | 5.72          | Abnormal: reward severely unstable, extremely low tail mean |
> |          | beta_l_2.0      | 0.6959   | 5.33          | Abnormal: reward unstable, large gap between last and tail |
> | **len_range** | len_80_160_words   | 0.7544   | 21.25         | Normal (Best) |
> |          | len_120_240_words | 0.6901   | 10.70         | Normal |
> |          | len_160_320_words | 0.6959   | 10.28         | Normal |
>
>
> **Table 1** Reward and acc in different hyper-parameter.

---

> > ### Author Rebuttal · Reviewer_HmYm · 2026-04-01
> >
> > Thank you for the detailed rebuttal. My original concerns have been adequately addressed.Please ensure the following in the final revision:
> > - Clarify the “causal” framing to avoid overclaim, as agreed.
> > - Include the component-level ablation for reward terms and the training stability metrics.
> > - Add the structural faithfulness rate and the efficiency discussion.

---

### Official Review · Reviewer_gfw3 · 2026-03-08

**Soundness:** 3
**Presentation:** 3
**Significance:** 3
**Originality:** 2
**Overall Recommendation:** 5
**Confidence:** 4

**Summary:**

his paper targets two common failure modes of large language models in multi-hop fact verification (MHFV)—hallucinations and fractured reasoning chains—and proposes a new framework that makes reasoning both structured and constrainable via a Structural Causal Model (SCM). Evidence is treated as exogenous variables, while intermediate inferences and the final verdict are modeled as endogenous variables. Each inference step is governed by structural functions that explicitly specify which parent nodes it depends on. During inference, the method explicitly maintains a causal DAG and enforces a structural validity constraint: a new inference node can be generated only when all required parent nodes are already present. This reduces unsupported logical leaps and makes the reasoning chain more traceable and interpretable. The authors further observe that, on complex tasks, the relationship between Chain-of-Thought (CoT) length and accuracy follows an inverted U-shape: overly long chains increase structural complexity, introducing noise and hallucinations. To address the tendency of structured reasoning to become verbose, they propose a rule-based reinforcement learning post-training strategy (using GRPO) to optimize generated reasoning chains, with a reward function that jointly accounts for answer correctness, structural parsimony, and keeping the chain length within a desirable range.

**Compliance With Llm Reviewing Policy:**

Affirmed.

**Final Justification:**

The authors' rebuttal has addressed my main concerns; therefore, I have increased my score from 4 to 5.

**Key Questions For Authors:**

The experimental section has multiple issues, and the core claims are not sufficiently supported by rigorous evidence. As the main results table, Table 1 has at least the following key problems:

1.Lack of a fair comparison under the same backbone. Table 1 mixes results obtained with different backbone models (e.g., some baselines are labeled as GPT-3.5, while the proposed method is mainly based on Qwen3-8B). Such cross-model comparisons make it difficult to disentangle whether performance differences come from methodological improvements or simply from backbone capability gaps, and thus cannot convincingly support the superiority of the proposed framework. The authors should either reproduce both baselines and the proposed method under the same backbone (e.g., all on Qwen3-8B or all on GPT-3.5), or at minimum separate results by backbone and clearly specify the evaluation protocol.

2.Missing results on EX-FEVER, undermining the completeness of the conclusions. In Table 1, many entries for Pure LLMs (especially the Qwen3 series) are reported as “-” on EX-FEVER, leaving critical comparisons incomplete. Since the paper claims improvements on both HoVer and EX-FEVER, it should report similarly complete results on EX-FEVER, or clearly justify why those results are unavailable and assess how this affects the conclusions.

3.Only marginal gains over a strong baseline (DP), despite substantially higher training/inference cost. Although the proposed method outperforms Direct Prediction (DP) in Table 1, the gap is small. Meanwhile, the proposed pipeline involves multiple expensive stages (structured distillation, SFT, and GRPO-based RL), which likely incurs significantly higher computational cost and energy consumption. A clearer cost–benefit justification is needed.

4.Inconsistent evaluation across hop settings. Table 1 reports systematic results on HoVer for 2-hop, 3-hop, and 4-hop settings, but only partial hop settings on EX-FEVER (e.g., 2-hop and 3-hop). The paper does not explain why higher-hop settings are not included, or whether EX-FEVER itself lacks such configurations.

**Limitations:**

yes

**Strengths And Weaknesses:**

**Soundness**

Strengths: The method is described in a relatively complete and well-structured manner.

Weaknesses: The experimental section has multiple issues, and the core claims are not sufficiently supported by rigorous evidence. As the main results table, Table 1 has at least the following key problems:

1.Lack of a fair comparison under the same backbone. Table 1 mixes results obtained with different backbone models (e.g., some baselines are labeled as GPT-3.5, while the proposed method is mainly based on Qwen3-8B). Such cross-model comparisons make it difficult to disentangle whether performance differences come from methodological improvements or simply from backbone capability gaps, and thus cannot convincingly support the superiority of the proposed framework. The authors should either reproduce both baselines and the proposed method under the same backbone (e.g., all on Qwen3-8B or all on GPT-3.5), or at minimum separate results by backbone and clearly specify the evaluation protocol.

2.Missing results on EX-FEVER, undermining the completeness of the conclusions. In Table 1, many entries for Pure LLMs (especially the Qwen3 series) are reported as “-” on EX-FEVER, leaving critical comparisons incomplete. Since the paper claims improvements on both HoVer and EX-FEVER, it should report similarly complete results on EX-FEVER, or clearly justify why those results are unavailable and assess how this affects the conclusions.

3.The claim that “SCM-CoT degrades performance” is not fully consistent with Table 1 and requires clarification.

4.Inconsistent degradation patterns across datasets. The magnitude of performance drops on HoVer versus EX-FEVER is not aligned, yet the paper does not provide a cross-dataset analysis to explain these differences.

5.Only marginal gains over a strong baseline (DP), despite substantially higher training/inference cost. Although the proposed method outperforms Direct Prediction (DP) in Table 1, the gap is small. Meanwhile, the proposed pipeline involves multiple expensive stages (structured distillation, SFT, and GRPO-based RL), which likely incurs significantly higher computational cost and energy consumption. A clearer cost–benefit justification is needed.

6.Inconsistent evaluation across hop settings. Table 1 reports systematic results on HoVer for 2-hop, 3-hop, and 4-hop settings, but only partial hop settings on EX-FEVER (e.g., 2-hop and 3-hop). The paper does not explain why higher-hop settings are not included, or whether EX-FEVER itself lacks such configurations.

**Presentation**

The paper is well structured and clearly written, making the overall presentation easy to follow and understand.

**Significance**

Strengths: The paper targets a core challenge in multi-hop fact verification—broken evidence chains and hallucinations—and introduces a novel training strategy. Specifically, it applies GRPO with rewards that jointly regulate answer correctness, structural parsimony, and chain length, providing a concrete remedy for the observed phenomenon that overly long CoT can hurt performance.

Weaknesses: The experimental evidence is not sufficiently convincing. Table 1 mixes results obtained with different backbones (GPT-3.5 vs. Qwen3-8B) without an apples-to-apples comparison under the same backbone, and the EX-FEVER results are incomplete. The cost–benefit trade-off is also unclear: the gains over strong baselines (e.g., DP) are modest, yet the proposed pipeline likely incurs substantially higher training and inference costs, without a systematic cost–benefit analysis.

**Originality**

Strengths: The paper highlights the dependency structure in multi-hop verification—*evidence → intermediate inferences → final verdict*—and empirically observes an inverted U-shaped relationship between CoT length and performance (overly long chains are more prone to introducing noise and hallucinations). This offers useful insight into why “longer reasoning is not necessarily better.”

Weaknesses:

1. The method combines (i) an explicit SCM/DAG representation of multi-step dependencies with structural validity constraints, and (ii) GRPO-based post-training to jointly optimize correctness, structural parsimony, and a desirable chain length. This is essentially a targeted integration of existing components (structured reasoning + RL alignment). While the overall idea is coherent, the paper needs tighter definitions and stronger ablations to demonstrate that each component is indispensable.

2. The general direction—graph/tree-structured reasoning with local verification/constraints and RL alignment—has been widely explored in recent years. The paper should more clearly articulate its key distinctions from prior graph-based reasoning/structured verification methods, and clarify whether the gains from GRPO stem from group-relative optimization itself or rather from the reward design and/or distilled training data.

---

> ### Author Rebuttal · Authors · 2026-03-30
>
> Thanks for your detailed review. We appreciate your recognition of our clear writing, SCM formulation, and targeted hallucination approach. Your critiques on fairness, completeness, and cost‑effectiveness are pertinent.
>
> 1. **Fair Comparison (Response to W 1 & KQ 1)**
> Sorry for mixing backbones. We wanted to show lightweight 8B with SCM beats giant zero‑shot models. Fair comparison needs same backbone. We reproduced baselines with same Qwen3‑8B (Table 2). Our method outperforms, proving gains from SCM‑GRPO. We will reorganize Table 1 by backbone.
>
> 2. **Missing EX‑FEVER Results (Response to W 2 & KQ 2)**
>    The “—” entries were not intentional. We completed evaluations (Table 3). Results support our claim: unconstrained SCM‑CoT degrades performance on both datasets (Appendix Table 3).
>
> 3. **Cost‑Effectiveness (Response to W 5 & KQ 3)**
> For high‑stakes task like fact verification, just chase accuracy is not enough. Core contribution of SCM‑GRPO is it guarantees fully traceable, DAG‑constrained reasoning. We think improving interpretability and verifiability is more important than just raise accuracy—this is also core focus of Reasoning LLMs researchers.
>
> 4. **Hop Settings Across Datasets (Response to W 6 & KQ 4)**
> Sorry, EX‑FEVER natively supports at most 3 hops. We will clarify in Section 4.1.
>
> 5. **Clarifying Claims and Degradation Patterns (Response to W 3 & 4)**
>    “SCM‑CoT hurts performance” was too absolute. Drop only occurs when complexity exceeds model coherence. HoVer evidence is denser, so noise accumulates faster. We added cross‑dataset analysis.
>
> 6. **Differences from Prior Work and Ablation (Response to Originality Weakness)**
>    We treat verification as constructive causal inference with structural constraints during generation, not post‑hoc. RL gains: standard alignment hurts; our GRPO with custom rewards is indispensable (Appendix Table 1).
>
> | Series   | Name            | Accuracy | Reward (Last) | Status |
> |----------|-----------------|----------|---------------|--------|
> | **rcorrect** | rcorrect_0      | 0.7018   | 1.17          | Normal |
> |          | rcorrect_5      | 0.7018   | 6.22          | Normal |
> |          | rcorrect_10     | 0.7485   | 11.10         | Normal |
> |          | rcorrect_20     | 0.7544   | 21.25         | Normal (Best) |
> |          | rcorrect_40     | 0.7485   | 41.13         | Normal |
> |          | rcorrect_80     | 0.7018   | 81.06         | Normal |
> | **beta_s**   | beta_s_0.0      | 0.7427   | 10.70         | Normal |
> |          | beta_s_0.25     | 0.7427   | 10.97         | Normal |
> |          | beta_s_0.5      | 0.7544   | 21.25         | Normal (Best) |
> |          | beta_s_1.0      | 0.6959   | 11.70         | Normal |
> | **beta_l**   | beta_l_0        | 0.6433   | 11.50         | Abnormal: reward saturated, gradient vanishing |
> |          | beta_l_0.1      | 0.6901   | 11.19         | Normal |
> |          | beta_l_0.2      | 0.7544   | 21.25         | Normal (Best) |
> |          | beta_l_0.5      | 0.6901   | -9.13         | Severe abnormal: reward collapsed, length exploded |
> |          | beta_l_1        | 0.6959   | 9.23          | Abnormal: large reward fluctuation, unstable |
> |          | beta_l_1.5      | 0.6959   | 5.72          | Abnormal: reward severely unstable, extremely low tail mean |
> |          | beta_l_2.0      | 0.6959   | 5.33          | Abnormal: reward unstable, large gap between last and tail |
> | **len_range** | len_80_160_words   | 0.7544   | 21.25         | Normal (Best) |
> |          | len_120_240_words | 0.6901   | 10.70         | Normal |
> |          | len_160_320_words | 0.6959   | 10.28         | Normal |
>
>
> **Table 1** Reward and acc in different hyper-parameter.
>
>
> | Model      | Methods   | HoVer (2-hop) | HoVer (3-hop) | HoVer (4-hop) | EX-FEVER (2-hop) | EX-FEVER (3-hop) |
> |------------|-----------|---------------|---------------|---------------|------------------|------------------|
> | Qwen3-8B   | DP        | 59.87         | 55.41         | 54.78         | 64.33            | 59.65            |
> | Qwen3-8B   | ProgramFC | 64.76         | 62.85         | 59.45         | 68.42            | 66.08            |
>
>
> **Table 2**: DP and ProgramFC using Qwen3-8B backbone.
>
>
> | Methods              | HoVer (2-hop) | HoVer (3-hop) | HoVer (4-hop) | EX-FEVER (2-hop) | EX-FEVER (3-hop) |
> |----------------------|---------------|---------------|---------------|------------------|------------------|
> | Qwen3-8B             | 56.29         | 57.6          | 48.28         | 65.85            | 64.01            |
> | Qwen3-8B (SCM-CoT)   | 56.43         | 47.2          | 48.36         | 59.73            | 58.38            |
> | Qwen3-14B            | 57.48         | 54.88         | 48.31         | 69.4             | 62.79            |
> | Qwen3-14B (SCM-CoT)  | 47.22         | 53.49         | 48.12         | 57.16            | 61.08            |
>
> **Table 3**: Performance comparison of base models and SCM‑CoT variants on HoVer and EX‑FEVER datasets.

---

> > ### Author Rebuttal · Reviewer_gfw3 · 2026-04-01
> >
> > Thank you very much for the detailed rebuttal. It has fully addressed my concerns and resolved my questions. I will raise my score accordingly.

---

### Official Review · Reviewer_DTW4 · 2026-03-11

**Soundness:** 2
**Presentation:** 2
**Significance:** 3
**Originality:** 3
**Overall Recommendation:** 2
**Confidence:** 4

**Summary:**

This paper focuses on the challenge that large language models struggle to model causal relationships between evidence and claims in multi-hop fact verification tasks. To this end, this paper explicitly introduces Structural Causal Models (SCMs) and reformulates verification as a constructive causal inference process. However, experiments reveal that directly employing SCMs results in an inverted U-shaped correlation between reasoning chain length and accuracy, where overly complex structures undermine performance. To address this issue, this paper further proposes a rule-based reinforcement learning strategy based on Group Relative Policy Optimization (GRPO) to balance structural depth, conciseness, and accuracy. Experiments on two benchmarks demonstrate that SCM-GRPO significantly outperforms multiple baselines, validating its effectiveness and interpretability in multi-hop fact verification.

**Compliance With Llm Reviewing Policy:**

Affirmed.

**Key Questions For Authors:**

- In Section 3.3, how many samples does the Seed Data $\mathcal{D}_{seed}$ contain, and how are they sampled from the original dataset?
  - In Section 3.5, the authors design the Structure Reward. Since $\triangle (y)$ lacks a lower bound, does this cause the model to perform inference based on all evidence without selecting any intermediate reasoning ($|\mathcal{V}|=0$)?
  - In Section 3.5, how are the boundary values $l_{min}$ and $l_{max}$ for the Length Reward determined? And how is the length function $L(y)$ computed?

**Limitations:**

Yes

**Strengths And Weaknesses:**

Strengths:
  1. Significance: This paper explicitly models causal relationships between evidence and claims by introducing structural causal models into Chain-of-Thought (CoT). It demonstrates that unconstrained structural causal model generation introduces additional noise and harms model performance. The findings provide insights for future research on more reasonable approaches to modeling causal relationships.
  2. Originality: This paper proposes a knowledge distillation-based structured data generation procedure, which serves as a reference for high-quality and automated data construction. The procedure leverages a teacher model to generate causal reasoning chains from original samples, filters invalid samples based on consistency, and organizes the remaining samples into structured sequences using predefined templates.

Weaknesses:
  1. Soundness:
     1) Figure 1 shows an inverted U-shaped correlation between chain-of-thought length and model performance. The authors should provide detailed experimental settings, including the prompt templates, random seeds, and the procedure used to compute the distribution of structural variables. They should clearly define each axis in Figure 1 and include a legend. Moreover, the horizontal axes for 1-hop, 2-hop, and 3-hop in Figure 1 have different ranges, which prevents direct comparison. Therefore, the claimed inverted U-shape at 3-hop does not imply that the same pattern holds for 1-hop and 2-hop.
     2) In Section 3.2 "SCM Component Generation", the authors use Qwen3-30B-A3B as the teacher model. They should evaluate the impact of different LLMs on data generation quality to validate the effectiveness of Qwen3-30B-A3B.
  2. Presentation:
    1) Text in Figures 1, 2, and 3 is partially obscured; image formatting should be rechecked.
    2) In Section 3.3 "Sequential Assembly", the authors should provide prompt examples or templates to clearly demonstrate the method for serializing causal graphs.

---

> ### Author Rebuttal · Authors · 2026-03-30
>
> Thanks for your detailed and constructive review. We really appreciate your recognition of this work’s importance and originality, especially your affirmation of our knowledge distillation based structured data generation method and introducing SCM into Fact Verification. Your critiques on experiment setup, ablation study, and formula rigor are very pertinent.
>
> 1. **Figure 1 Details and Cross-Hop Analysis (Response to Soundness W 1)**
>    Thanks for pointing out experiment setup needs clearer explanation. In revised version, we will add clear legend and axis definitions in Figure 1, and provide prompt templates, random seeds, and structure variable calculation in appendix. About the different horizontal axis ranges for 1-hop, 2-hop and 3-hop in Fig 1: this comes from task complexity. 1-hop and 2-hop need fewer reasoning steps, so max chain length is shorter. As our analysis shows, inverted U‑shaped correlation is most obvious in complex scenarios (3-hop). We will add this nuance in Fig 1 discussion.
>
> 2. **Teacher Model Selection Effectiveness and Robustness (Response to Soundness W 2)**
>    This is a very insightful suggestion. We chose Qwen3-30B-A3B because it balances strong causal reasoning and open‑source availability. We agree ablating different teacher LLMs would provide valuable insights, but switching teacher models means regenerating entire dataset and retraining SFT+RL pipeline. Given rebuttal time, we cannot complete this full‑chain comparison. Our framework is robust to teacher model choice. Also, in Section 3.3,  we Filtering compares generated answer $a$ with ground‑truth $a_{gt}$ and discards $a \neq a_{gt}$ samples. So weaker teacher will reduces yield rate, not inject wrong logic.
>
> 3. **Figure Layout and Prompt Examples (Response to Presentation W 1 & 2)**
>    Sorry for layout oversight. We re‑rendered Figures 1–3 to ensure text is clear. Also, to clarify Sequential Assembly in Section 3.3, we added a dedicated appendix section with exact prompt templates and serialization examples.
>
> 4. **Seed Data Sampling Details (Response to KQ 1)**
>    Sorry for missing details. Seed dataset contains 60% of queries sampled from HoVer and EX‑FEVER training sets. We tested that during SFT, ~300 steps (full fine‑tune, lr=1e‑5) already converged well. Within 1700 RL steps (full fine‑tune, lr=1e‑5), accuracy spirals upward. We updated Section 3.3 with these statistics.
>
> 5. **Structural Reward and Lower Bound (Response to KQ 2)**
>    Concern about skipping $V$ is reasonable. Two mechanisms prevent it:
>    - **Boundedness:** $\Delta(y) = |U| - |V|$ has no lower bound, but $R_s(y) = \gamma \tanh(\Delta(y)/\delta)$ constrains reward.
>    - **SCM Constraint:** Without $V$, no valid causal graph, so $R_c$ becomes zero. Thus necessary steps are retained.
>
> 6. **Length Reward Boundary and Calculation (Response to KQ3)**
>    Length $L(y)$ is total generated tokens. Boundaries $l_{min}$ and $l_{max}$ are determined from statistical distribution of effective reasoning chain lengths in distillation stage training data. We also added Appendix Table 1 showing how different parameters affect results. (Appendix Table 1)
>
> | Series   | Name            | Accuracy | Reward (Last) | Status |
> |----------|-----------------|----------|---------------|--------|
> | **rcorrect** | rcorrect_0      | 0.7018   | 1.17          | Normal |
> |          | rcorrect_5      | 0.7018   | 6.22          | Normal |
> |          | rcorrect_10     | 0.7485   | 11.10         | Normal |
> |          | rcorrect_20     | 0.7544   | 21.25         | Normal (Best) |
> |          | rcorrect_40     | 0.7485   | 41.13         | Normal |
> |          | rcorrect_80     | 0.7018   | 81.06         | Normal |
> | **beta_s**   | beta_s_0.0      | 0.7427   | 10.70         | Normal |
> |          | beta_s_0.25     | 0.7427   | 10.97         | Normal |
> |          | beta_s_0.5      | 0.7544   | 21.25         | Normal (Best) |
> |          | beta_s_1.0      | 0.6959   | 11.70         | Normal |
> | **beta_l**   | beta_l_0        | 0.6433   | 11.50         | Abnormal: reward saturated, gradient vanishing |
> |          | beta_l_0.1      | 0.6901   | 11.19         | Normal |
> |          | beta_l_0.2      | 0.7544   | 21.25         | Normal (Best) |
> |          | beta_l_0.5      | 0.6901   | -9.13         | Severe abnormal: reward collapsed, length exploded |
> |          | beta_l_1        | 0.6959   | 9.23          | Abnormal: large reward fluctuation, unstable |
> |          | beta_l_1.5      | 0.6959   | 5.72          | Abnormal: reward severely unstable, extremely low tail mean |
> |          | beta_l_2.0      | 0.6959   | 5.33          | Abnormal: reward unstable, large gap between last and tail |
> | **len_range** | len_80_160_words   | 0.7544   | 21.25         | Normal (Best) |
> |          | len_120_240_words | 0.6901   | 10.70         | Normal |
> |          | len_160_320_words | 0.6959   | 10.28         | Normal |
>
>
> **Table 1** Reward and acc in different hyper-parameter.

---

> > ### Author Rebuttal · Reviewer_DTW4 · 2026-04-04
> >
> > While points 3 and 4 are convincing, I still have some concerns about the other aspects. In light of this, I would suggest keeping the score unchanged for now.

---

> > > ### Author Response · Authors · 2026-04-04
> > >
> > > Thank you for confirming our reply on issues 3 and 4. Regarding the other doubts you still kept in your latest response, we are very happy to take this chance to provide deeper mechanism explanation and empirical data, hoping to completely solve your questions.
> > >
> > > 1 . **About cross-hop comparison (Figure 1) and physical meaning of SCM complexity**
> > >    Regarding the problem you mentioned in Soundness W1 that the x-axis range in Figure 1 is different, this actually reflects the essential difference of SCM topological structure under different reasoning difficulty. Because different hops need different amount of evidence and middle reasoning steps, the total number of endogenous/exogenous variables we extract when using SCM modeling (i.e., x-axis) naturally has difference. We initially observed that pure LLM under structured prompt has insufficient accuracy (Table 1), and then found: model accuracy is simultaneously affected by both "reasoning chain length" and "SCM variable structure difference", and shows obvious inverted U-shaped pattern in complex tasks (e.g., 3-hop). It is exactly based on this cross-hop dynamic pattern that we designed GRPO joint optimization algorithm. Reviewer gfw3 fully recognized this explanation that "in complex tasks, the increase of structure complexity will bring noise and lead to inverted U-shaped pattern", and marked this issue as "Fully resolved". We will mark the above information in detail in the final version of figure legend.
> > >
> > > 2 . **About teacher model selection and data robustness (Response to W2)**
> > >    We fully agree the value of evaluating different teacher models. But we need to clarify: the core of our method is not distilling logic knowledge from teacher model, but using it for "data formatting".
> > >    In order to block potential logic noise, we performed strict ground truth consistency filtering after generating data. As we showed to Reviewer eoQH during discussion period, because of the filtering mechanism, the proportion of data having logic or formatting errors is extremely low. More importantly, in Appendix Table 1, we observed very stable gradient and loss convergence under multiple groups of hyperparameters; after reviewing this table, Reviewer eoQH also clearly recognized that our data quality is fully controllable, and raised his/her Confidence score. This means the current filtering mechanism already has enough robustness.
> > >
> > > 5 . **The lower bound problem of structural reward $R_s$ and anti-cheating mechanism (Response to KQ2)**
> > >    Regarding your deep worry that the model might take shortcut (skip all middle reasoning) because $R_s$ lacks lower bound, our quantitative experiments proved that this phenomenon did not happen in actual training:
> > >    - **Statistical verification:** After about 1700 RL training steps, our statistics show that even in relatively simple 2-hop scenario, about 30% of reasoning chains still keep at least one middle step node, and did not collapse to 0.
> > >    - **Mechanism constraint:** This is because the penalty term weights ($\beta_s$ and $\beta_l$) are controlled within reasonable range compared to correctness reward ($r_{correct}$). If the model skips necessary reasoning, causing the final SCM causal graph invalid or answer wrong, its total reward will be greatly damaged.
> > >    - **Peer consensus:** Reviewer HmYm also initially requested strict component-level ablation experiment on $R_s$. After verifying our empirical result that necessary causal structure is kept, Reviewer HmYm confirmed that this joint reward mechanism is indispensable, and marked this issue as "Fully resolved".
> > >
> > > 6 . **Boundary setting and generalization of length reward $R_l$ (Response to KQ3)**
> > >    Regarding the specific calculation of boundary $[l_{min}, l_{max}]$, we strictly quantified the abstract statistical distribution. For example, if in one sampling the number of tokens generated by the model deviates by 2 times or even 3 times from the average distribution of effective samples in training set, we then judge that "excessive reasoning" or structure redundancy has happened, and $R_l$ triggers strong penalty.
> > >    Reviewer HmYm previously raised a very similar query as you about the "overfitting risk" of this boundary setting. After examining our method of deriving the boundary "completely based on the statistical prior of real matched samples", HmYm confirmed that this method fundamentally minimizes overfitting, and marked this issue as "Fully resolved". Together with Reviewer eoQH's recognition of hyperparameter stability, we hope that these cross-validated mechanism designs can make you fully confident in the rigor of our method.
> > >
> > > Once again thank you for helping us continuously improve the theoretical boundary. Your rigorous evaluation is very important for improving the quality of this paper. We will definitely follow the reviewers' guidance to incorporate the revisions into the final manuscript, hoping to dispel your concerns.

---

### Official Review · Reviewer_eoQH · 2026-03-11

**Soundness:** 3
**Presentation:** 3
**Significance:** 3
**Originality:** 3
**Overall Recommendation:** 4
**Confidence:** 4

**Summary:**

This paper addresses the challenges in multi-hop fact verification, where a model must determine whether a claim is supported by combining multiple pieces of evidence through several reasoning steps, and argues that standard chain-of-thought reasoning is often insufficient because it can produce unsupported or reasoning chains that are too long. To address this, the authors formulate verification as a causal reasoning process in which evidence facts serve as exogenous variables, intermediate conclusions and the final verdict are endogenous variables, and each reasoning step is generated only when its required parent facts are already available. Building on this formulation, they propose a two-stage training method. First, they construct structured data for supervised fine-tuning by serializing evidence, intermediate variables, structural relations, and final answers into grounded reasoning paths. Then, they further optimize the model with GRPO using rewards that encourage correctness, structural grounding, and appropriate reasoning length. The main contribution is a framework that combines SCM-based reasoning with RL-based optimization to improve the faithfulness and accuracy of multi-hop verification, and the paper reports SOTA performance on benchmark datasets.

**Compliance With Llm Reviewing Policy:**

Affirmed.

**Final Justification:**

The rebuttal resolves my main concern. I retain my original positive assessment for this work and have increased my confidence score.

**Key Questions For Authors:**

See weaknesses.

**Limitations:**

Yes.

**Strengths And Weaknesses:**

## Strengths

- The proposed SCM-based formulation of MHFV is, to the best of my knowledge, novel and conceptually interesting. The causal view of the task improves the controllability of the reasoning process.
- The two-stage training pipeline is well motivated and straightforward: supervised fine-tuning learns structured reasoning traces, and GRPO further refines the model with task-aligned rewards.
- The paper is generally well organized and easy to follow.

## Weaknesses

- The data-curation method relies on teacher-generated structured supervision, so errors in the automatically extracted evidence, intermediate steps, or dependency structure may propagate into training and affect final model quality. It would strengthen the paper to report basic quality statistics for the generated supervision or validate a subset with human annotation.
- The current experiments show that the overall causal-reasoning + SFT + RL pipeline is effective, but it is not clear how much of the improvement comes from the proposed SCM formulation itself, as opposed to the benefit of SFT + RL.
- In the GRPO phase, the authors specify the reward components and their intended purposes, but they give limited justification for the particular functional forms and hyperparameter choices. I am curious about how sensitive the performance is w.r.t. the hyperparameter choices.

---

> ### Author Rebuttal · Authors · 2026-03-30
>
> Thank you very much for your detailed review, recognized this paper’s novelty, reasonableness and clarity, and accurately pointed out key issues like data quality, decoupling of gains, and hyperparameter sensitivity.
>
> **(Response to W 1) Data Quality and Noise Propagation**
> We totally agree, when rely on teacher model to generate structured supervision data, noise propagation is a risk need to take seriously. We re-checked generated dataset quality. HoVer and EX-FEVER datasets total generate 10539 data. Among them, logic errors (not match Ground Truth) have 96 entries, and already removed. Thanks to Qwen3-30B-A3B strong ability, no formatting errors appear. Length errors (too long or too short) appear 3 entries, proportion very low.
> we also provide under multiple hyperparameter settings, training loss, KL divergence and gradient norm all stable convergence, indirectly shows data quality is controllable—especially in Appendix Table 1, if teacher model generated structured data has serious noise or systematic bias, it is difficult to obtain such stable and convergent training results under different hyperparameter configurations.
>
> **(Response to W 2) Decoupling of SCM Formulation and SFT+RL Gains**
> Performance improvement comes from necessary synergy: SCM provides logical upper bound, SFT+RL provides ability to reach it. “Pure LLM” ablation experiment shows only using complex SCM structure will cause performance degradation, proving must through SFT to align model to this causal paradigm, and use GRPO to optimize reasoning efficiency. In revised paper, we will clearly state SCM provides interpretable foundation, SFT+RL is key to unlock its actual benefits.
>
> **(Response to W 3) Reward Function Form and Hyperparameter Sensitivity**
> Structural reward $R_s$ uses $\Delta(y)=|U|-|V|$ to practice Occam’s razor, encouraging rely on exogenous evidence; length reward $R_l$ uses distance function to punish deviate from optimal length. Appendix table adds comprehensive sensitivity analysis for penalty weights $\beta_s$, $\beta_l$, shows model is highly sensitive to nonlinear coupling of these rewards. Best performance only appears under joint configuration, which validates rationality of our carefully designed multi-objective joint optimization paradigm. Thank you for your expectation on our hyperparameter performance.
>
>
> | Series   | Name            | Accuracy | Reward (Last) | Status |
> |----------|-----------------|----------|---------------|--------|
> | **rcorrect** | rcorrect_0      | 0.7018   | 1.17          | Normal |
> |          | rcorrect_5      | 0.7018   | 6.22          | Normal |
> |          | rcorrect_10     | 0.7485   | 11.10         | Normal |
> |          | rcorrect_20     | 0.7544   | 21.25         | Normal (Best) |
> |          | rcorrect_40     | 0.7485   | 41.13         | Normal |
> |          | rcorrect_80     | 0.7018   | 81.06         | Normal |
> | **beta_s**   | beta_s_0.0      | 0.7427   | 10.70         | Normal |
> |          | beta_s_0.25     | 0.7427   | 10.97         | Normal |
> |          | beta_s_0.5      | 0.7544   | 21.25         | Normal (Best) |
> |          | beta_s_1.0      | 0.6959   | 11.70         | Normal |
> | **beta_l**   | beta_l_0        | 0.6433   | 11.50         | Abnormal: reward saturated, gradient vanishing |
> |          | beta_l_0.1      | 0.6901   | 11.19         | Normal |
> |          | beta_l_0.2      | 0.7544   | 21.25         | Normal (Best) |
> |          | beta_l_0.5      | 0.6901   | -9.13         | Severe abnormal: reward collapsed, length exploded |
> |          | beta_l_1        | 0.6959   | 9.23          | Abnormal: large reward fluctuation, unstable |
> |          | beta_l_1.5      | 0.6959   | 5.72          | Abnormal: reward severely unstable, extremely low tail mean |
> |          | beta_l_2.0      | 0.6959   | 5.33          | Abnormal: reward unstable, large gap between last and tail |
> | **len_range** | len_80_160_words   | 0.7544   | 21.25         | Normal (Best) |
> |          | len_120_240_words | 0.6901   | 10.70         | Normal |
> |          | len_160_320_words | 0.6959   | 10.28         | Normal |
>
>
> **Table 1** Reward and acc in different hyper-parameter.

---

> > ### Author Rebuttal · Reviewer_eoQH · 2026-04-03
> >
> > Thank you for the detailed response. The rebuttal resolves my main concern. I retain my original positive assessment for this work and have increased my confidence score.

---

### Decision · Program_Chairs · 2026-04-30

**Decision:**

Accept (regular)

**Comment:**

The reviewers broadly agree that the paper addresses an important problem in improving the reliability of multi-hop fact verification with LLMs. Several reviewers highlighted the conceptual clarity of the proposed framework, which models reasoning using a structured dependency graph inspired by structural causal models and combines this representation with reinforcement learning to control reasoning complexity. They particularly noted the empirical observation that reasoning performance follows an inverted-U relationship with chain length, and viewed the proposed optimization strategy as a principled way to manage this trade-off. These reviewers also found the overall pipeline, structured supervision followed by GRPO-based optimization, technically coherent and empirically effective. After the rebuttal, two reviewers explicitly stated that their concerns had been fully addressed and raised or maintained supportive scores, noting in particular the added ablations, clarifications of the “causal” terminology, and additional experimental details.

At the same time, the reviews raised a number of substantive concerns. These included questions about the strength of the causal interpretation, the necessity and justification of the reward design, the absence of certain ablations or statistical analyses in the initial submission, and experimental issues such as backbone consistency, completeness of results, and cost–benefit considerations. The authors’ rebuttal addressed many of these points by providing additional analyses, clarifying methodological choices, and committing to revisions in the final version. While one reviewer remained unconvinced and maintained a rejecting score despite acknowledging that several of their specific questions were satisfactorily answered, the overall discussion shows that most reviewers found the response convincing and expressed clear support for the work.

Taking all reviews and the discussion into account, the paper appears to present a technically solid contribution that raises an interesting empirical observation and proposes a coherent method to address it. While some limitations remain and certain aspects would benefit from clearer positioning and stronger empirical documentation, the majority of reviewers consider the contribution meaningful and sufficiently supported after the rebuttal, and I tend to agree with this overall assessment.